# ROBOPARA: DUAL-ARM ROBOT PLANNING WITH PARALLEL ALLOCATION AND RECOMPOSITION ACROSS TASKS

**Shiying Duan**[1,*], **Pei Ren**[2,*,‡], **Nanxiang Jiang**[1], **Zhengping Che**[2], **Jian Tang**[2],
**Zhaoxin Fan**[1,†], **Yifan Sun**[3,†], **Wenjun Wu**[1]

[1]Beijing Advanced Innovation Center for Future Blockchain and Privacy Computing,
School of Artificial Intelligence, Beihang University
[2]Beijing Innovation Center of Humanoid Robotics
[3]Center for Applied Statistics, School of Statistics, Renmin University of China

[*]Equal contribution.    [†]Corresponding authors.    [‡]Project leader.

## ABSTRACT

Dual-arm robots play a crucial role in improving efficiency and flexibility in complex multitasking scenarios. While existing methods have achieved promising results in task planning, they often fail to fully optimize task parallelism, limiting the potential of dual-arm collaboration. To address this issue, we propose RoboPARA, a novel large language model (LLM)-driven framework for dual-arm task parallelism planning. RoboPARA employs a two-stage process: (1) Dependency Graph-based Planning Candidates Generation, which constructs directed acyclic graphs (DAGs) to model task dependencies and eliminate redundancy, and (2) Graph Re-Traversal-based Dual-Arm Parallel Planning, which optimizes DAG traversal to maximize parallelism while maintaining task coherence. In addition, we introduce the Cross-Scenario Dual-Arm Parallel Task dataset (X-DAPT dataset), the first dataset specifically designed to evaluate dual-arm task parallelism across diverse scenarios and difficulty levels. Extensive experiments demonstrate that RoboPARA significantly outperforms existing planning methods, achieving higher efficiency and reliability, particularly in complex task combinations. Our code is publicly available at https://github.com/AiDuanshiying/RoboPARA.

## 1 INTRODUCTION

Dual-arm robot manipulation represents a transformative milestone in embodied AI (Liu et al., 2025a), offering unprecedented capabilities to tackle complex and collaborative tasks (Kurosu et al., 2017; Gao & Yu, 2022; Takata et al., 2022). By leveraging coordinated actions, dual-arm systems overcome the sequential limitations of single-arm robots and significantly enhance both efficiency and precision (Smith et al., 2012; Gao & Yu, 2022). Central to unlocking these advantages is effective task planning (Guo et al., 2023), which bridges high-level objectives with seamless execution. Recently, the rise of large language models (LLMs) has further advanced dual-arm task planning, providing innovative approaches to improve coordination, adaptability, and overall system performance (Huang et al., 2024; Ruan et al., 2023; Sharan et al., 2023).

Indeed, a growing body of research has been proposed that leverages LLMs for task planning in dual-arm robot manipulation (Ahn et al., 2022; Bai et al., 2025; Wei et al., 2023; Mu et al., 2023). For example, RoCo (Mandi et al., 2023) prompts LLM agents to iteratively improve their plans and waypoints in-context, enabling more refined task execution strategies. Similarly, FLTRNN (Zhang et al., 2024) employs a language-based RNN structure to integrate task decomposition and memory management into the LLM planning inference process. Despite their innovative designs, these methods often result in single-arm sequential execution. This is because they primarily focus on optimizing task success rate and completion time, while neglecting a critical factor: parallelism between the two arms. As a result, the collaborative potential of dual-arm systems is not fully realized.

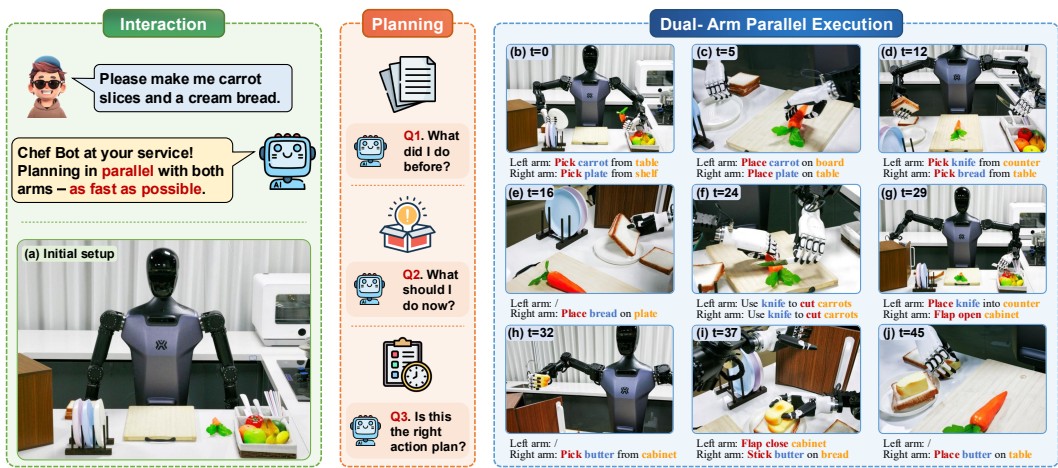

Figure 1: RoboPARA enables efficient parallel manipulation and collaborative dual-arm execution, resulting in a 30% to 50% reduction in execution time while maintaining task success. When deployed on a humanoid robot, it demonstrates behaviors closely aligned with human activities.

Specifically, in real-world applications, multiple tasks are often issued concurrently, posing a fundamental challenge: how to plan efficiently under multitasking demands. To address this, we draw inspiration from everyday human behavior. Consider a typical morning routine, actions such as boiling water, brushing teeth, and washing your face are naturally interleaved—for example, you may let the water boil while brushing your teeth. This ability to reason about temporal overlap and parallel execution is key to human efficiency. Similarly, in a robotic kitchen, the same principle applies. Some tasks require both arms to work together, such as cutting carrots, while others, like chopping vegetables, only require one arm, leaving the other free to perform a different task. Deciding when to synchronize both arms and when to decouple them becomes crucial for optimizing execution time. Despite its practical importance, this dimension of parallelism-aware dual-arm planning remains largely underexplored in current research.

To tackle the issue, we propose RoboPARA, a novel LLM-driven dual-arm robot planning framework. RoboPARA ensures both task correctness and optimal arm utilization through a two-stage architecture: **Dependency Graph-based Planning Candidates Generation** and **Graph Re-Traversal-based Dual-Arm Parallel Planning**. In the former stage, RoboPARA processes user instructions by utilizing a local memory module, supported by retrieval augmented generation (RAG) (Gao et al., 2023), to obtain detailed procedural knowledge relevant to the task. This retrieved information is integrated into a structured prompt, enabling the LLM to generate a directed acyclic graph (DAG) with correction that outlines the step dependencies. In the later stage, the generated DAG is analyzed and optimized by scheduling algorithms to fully exploit the collaborative potential of dual-arm robots. Together, these two stages form a tightly integrated pipeline that enables RoboPARA to generate task plans that not only ensure accuracy and feasibility but also optimize dual-arm collaboration.

To validate the effectiveness of RoboPARA, we further propose the Cross-Scenario **D**ual-**A**rm **P**arallel **T**ask dataset (X-DAPT dataset), the first dataset designed to evaluate dual-arm task planning parallelism. Our dataset includes more than 1,000 task packages across 10 key scenarios, each divided into three difficulty levels. Experiments demonstrate that RoboPARA achieves outstanding performance, especially in the most complex task combinations. Compared to existing methods, RoboPARA exhibits over $4.5\times$ parallel and collaborative steps in average, resulting in a 30% to 50% reduction in execution time, significantly improving efficiency. Moreover, RoboPARA achieves 34% higher success rate than the average of other methods in challenging tasks, ensuring greater reliability and stability in task execution.

Our contributions are summarized as follows:

- **New Task**: We propose a new dual-arm task planning problem, the **Dual-Arm Cooperative Scheduling Problem**, specifically designed to optimize parallelism and improve multitasking efficiency in real-world scenarios. Our work establishes a new task and objective, pushing the boundaries of dual-arm robotic manipulation.

- **New Dataset**: We design the **X-DAPT dataset**, the first dataset tailored for evaluating dual-arm task parallelism. It includes over 1,000 task packages across 10 diverse scenarios with difficulty levels, offering a comprehensive benchmark.

- **New Method**: We propose **RoboPARA**, an LLM-driven dual-arm planning framework using a two-stage process: Dependency Graph-based Planning and Graph Re-Traversal. It ensures efficient, reliable, and highly parallel task execution, achieving state-of-the-art task success rates and execution efficiency on X-DAPT dataset and real-world scenarios.

## 2 RELATED WORK

**Dual-arm Robot Manipulation.** Dual-arm manipulation plays a critical role in enabling robots to perform complex, high-precision tasks in various domains (Zhao et al., 2025; Yoshida et al., 2023). Current approaches to dual-arm manipulation can be broadly categorized into two main paradigms: End-to-end methods train a self-contained model to directly map from task descriptions to robotic trajectory, such as HybridVLA (Liu et al., 2025b), DA-VIL (Karim et al., 2024), R3M (Nair et al., 2022), Long-VLA (Fan et al., 2025). However, such methods have limited scalability and flexibility for long-horizon tasks, and generalizing to new tasks or environments often requires extensive retraining. Compositional skill learning involves developing multiple meta-skills that can be combined to accomplish complex tasks, offering greater flexibility and reusability (Gu et al., 2023; Villagrossi et al., 2021; Gu et al., 2022). Approaches like RoboTwin (Mu et al., 2025), RoboCodeX (Mu et al., 2024), and AnyGrasp (Fang et al., 2023) leverage modular representations of skills to enable dynamic recomposition in novel settings. This approach is well-suited for long-horizon tasks. Our work is built on this compositional paradigm to leverage its generalization capability.

**Large Language Models for Task Planning.** LLMs have demonstrated remarkable capabilities in robotic task planning, translating high-level tasks into actionable steps and enabling more adaptive, coordinated behaviors (Bai et al., 2025; Mandi et al., 2023; Zhao et al., 2025). Key planning methods include direct task decomposition and structured coordination. Direct task decomposition use LLMs to break down complex instructions into linear or hierarchical sequences of actions (Wake et al., 2023; Wu et al., 2023; Huang et al., 2022; Raman et al., 2024; Xie et al., 2023). Examples include LLM-Planner (Song et al., 2023) and DELTA (Liu et al., 2025c), where tasks are structured step-by-step for easier execution. Structured coordination solutions further focus on exploiting the parallel capabilities of dual-arm or multi-agent systems (Bai et al., 2025; Zhao et al., 2025; Khan et al., 2025), these approaches are designed to promote parallelism through the coordination of concurrent actions. Approaches such as LLM+MAP (Chu et al., 2025), Tree-Planner (Hu et al., 2024), RoCo (Mandi et al., 2023), and emerging graph-based planners (Byeon & Oh, 2024; Zhang et al., 2025) fall into this category, enabling structured modeling of task dependencies and parallel execution potential.

**System-1 + System-2 Framework in Embodied AI.** Embodied algorithms generally follow two paradigms: end-to-end methods (Liu et al., 2025b; Karim et al., 2024; Nair et al., 2022), which directly map task descriptions to trajectories but struggle with long-horizon generalization, and the System-1 + System-2 hierarchical framework (Stanovich et al., 2000; Kahneman, 2011; Liu et al., 2025a), where System-2 handles symbolic reasoning and scheduling while System-1 executes low-level actions. The latter has shown greater practicality for complex multi-arm settings (Holladay et al., 2024; Adu-Bredu et al., 2022; Vu et al., 2024; Joublin et al., 2024). Nevertheless, high-level System-2 planning for long-horizon, parallel dual-arm collaboration remains underexplored. Our work specifically addresses this gap by enhancing System-2 planning to improve efficiency, parallelism, and adaptability in dynamic environments.

## 3 METHODOLOGY

### 3.1 PROBLEM FORMULATION

We focus on the System-2 component within the well recognized System-1 + System-2 architecture, and formalize the dual-arm collaborative scheduling problem as a dual-arm multi-task optimization problem with temporal and structural constraints. Let $\mathcal{P} = \{P_1, P_2, \ldots, P_m\}$ denote a collection of task packages, where each $P_i$ is composed of a sequence of meta operations.

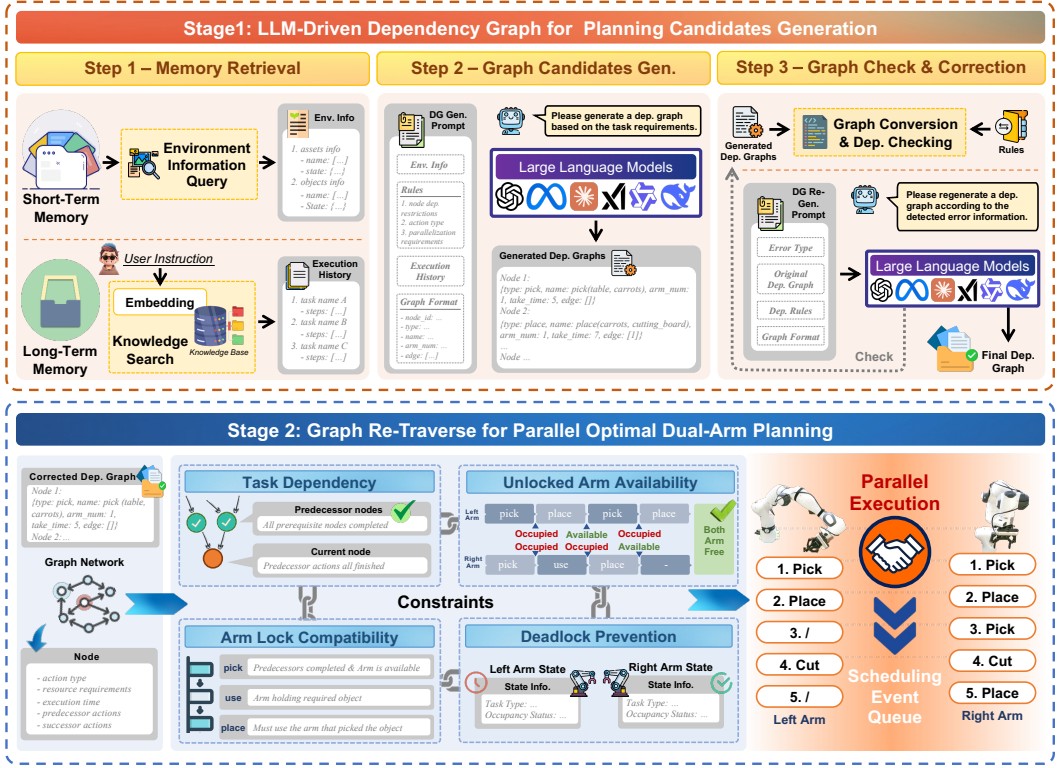

Figure 2: The RoboPARA framework. Given user instructions, our two-stage framework outputs a dual-arm execution schedule. Stage 1 generates dependency graph-based planning candidates: it builds an LLM-based DAG from experiences and iteratively refines it via error correction (Appendix Alg. 2) and updates. Stage 2 re-traverses the graph to plan parallel execution: it identifies parallelizable tasks and resolves deadlocks (Appendix Alg. 4 and Alg. 3), then finalizes task assignment.

All operations from all packages are consolidated into a global DAG $\mathcal{G} = (\mathcal{V}, \mathcal{E})$, where each node $v \in \mathcal{V}$ corresponds to a meta operation, encoded as $(\texttt{name}_v, \texttt{type}_v, \delta_v, t_v)$, which defines the action label, semantic type, required number of arms $\delta_v \in \{1, 2\}$, and execution duration $t_v \in \mathbb{R}^+$. Each directed edge $(v_i \rightarrow v_j) \in \mathcal{E}$ denotes that $v_j$ cannot start before before $v_i$ completes.

Let $\mathcal{A} = \{L, R\}$ be the set of available robotic arms. We define an arm assignment function:

$$\mu : \mathcal{V} \rightarrow \mathcal{P}(\mathcal{A}), \quad \text{with } |\mu(v)| = \delta_v, \tag{1}$$

which assigns each operation to its arm(s), respecting exclusivity and synchronization semantics.

We further define a scheduling function $\sigma : \mathcal{V} \rightarrow \mathbb{R}^+$, assigning each node a non-negative start time. The pair $(\sigma, \mu)$ must satisfy all task dependencies, arm availability constraints, and consistency requirements throughout execution.

The effectiveness of a dual-arm execution plan is measured by its *makespan*—the maximum completion time of all operations:

$$C_{\max} := \max_{v \in \mathcal{V}} (\sigma(v) + t_v). \tag{2}$$

Our objective is to find a valid schedule and arm assignment that jointly minimize this makespan:

$$\min_{\sigma, \mu} C_{\max}, \quad \text{subject to } (\sigma, \mu) \in \mathcal{F}(\mathcal{G}, \mathcal{S}), \tag{3}$$

where $\mathcal{S}$ denotes the environment state, and $\mathcal{F}(\mathcal{G}, \mathcal{S})$ denotes the feasible set of schedule-allocation pairs satisfying structural, temporal, and contextual constraints.

## 3.2 Overall Framework

RoboPARA tackles the newly defined **Dual-Arm Cooperative Scheduling Problem** through a two-stage architecture: **Dependency Graph-based Planning Candidate Generation** and **Graph Re-Traversal-based Dual-Arm Parallel Planning**. The overall framework is illustrated in Fig. 2. The first stage focuses on generating multiple planning candidates, while the second stage performs efficient parallel scheduling, selecting the optimal execution plan. The final output is a complete, time-synchronized execution plan detailing meta operations, durations, and arm assignments, ensuring task correctness and efficient arm utilization. The following sections elaborate on each stage of RoboPARA, detailing the core algorithmic components and innovations underlying our design.

## 3.3 LLM-Driven Dependency Graph for Planning Candidates Generation

This stage constructs the task-level DAG $\mathcal{G} = (\mathcal{V}, \mathcal{E})$, serving as the structural foundation for dual-arm scheduling. Given a user instruction, RoboPARA employs RAG to fetch relevant task knowledge from a hybrid memory system. This memory integrates both short-term observations (*e.g.*, object states) and long-term execution history (*e.g.*, package knowledge base).

The retrieved task steps are then augmented with additional contextual information, such as environmental constraints, dependency rules, parallelism guidelines, and structured formatting examples, to form a full prompt (Appendix Sec. A.2.2). The prompt is given to an LLM to produces an initial instance of $\mathcal{G}$, where each node $v \in \mathcal{V}$ is categorized into operation types such as `pick`, `use`, `place`, `open`/`close` and `complete`. The dependency edges $\mathcal{E}$ reflect precedence constraints—for example, a `use` action must follow a corresponding `pick` of the same tool, and a `place` must not precede the associated `use`. Additionally, tool-related action sequences must maintain intra-object dependency consistency. The `complete` node serves as the unique sink of $\mathcal{G}$, aggregating all terminal dependencies to represent the logical end of the task sequence.

Since the initial $\mathcal{G}$ may contain errors in dependency relationships, we apply a structural validation routine (Appendix Alg. 2) to detect and localize errors. Three key categories of invalid dependencies are identified: (1) an operation (`use` or `place`) incorrectly depending on another object's `place`; (2) a `place` node in a `pick-use-place` sequence directly depends on its `pick` node instead of the preceding `use` node; (3) any node depending on the `use` node of an unrelated other object.

Once errors are identified, related nodes and dependency rules are integrated into a new prompt for iterative correction. The updated prompt is re-submitted to the LLM to regenerate a valid $\mathcal{G}$. After validation, the graph $\mathcal{G}$ proceeds to scheduling and execution planning.

## 3.4 Graph Re-Traverse for Parallel Optimal Dual-Arm Planning

Given a validated task graph $\mathcal{G} = (\mathcal{V}, \mathcal{E})$, RoboPARA constructs a feasible and efficient dual-arm execution plan $(\sigma, \mu)$, assigning each node $v \in \mathcal{V}$ a start time $\sigma(v)$ and a responsible arm set $\mu(v) \subseteq \mathcal{A}$, minimizing makespan $C_{\max}$ while respecting all structural and physical constraints.

**Scheduling Framework.** We maintain a dynamic scheduling queue $\mathcal{Q}$ with all nodes $v$ that satisfy:
$$\text{Ready}(v) := \{v \in \mathcal{V} \mid \text{ all predecessors } u \in \text{pred}(v) \text{ are scheduled}\} \qquad (4)$$
Each node in $\mathcal{Q}$ is assigned based on arm availability, task type ($\delta_v = 1$ or $2$), and object-holding consistency. Single-arm tasks go to the idle arm, favoring the left in case of a tie. Dual-arm tasks require both arms free and not object-locked.

**Formal Constraints.** To ensure correctness and feasibility, the plan $(\sigma, \mu)$ must satisfy the following four categories of constraints:

1. **Task Dependency.** Each task $v \in \mathcal{V}$ can only be scheduled after the completion of all its predecessors. Formally:
$$\sigma(v) \geq \max_{u \in \text{pred}(v)} (\sigma(u) + t_u), \qquad (5)$$
   where $\text{pred}(v) := \{u \in \mathcal{V} \mid (u \to v) \in \mathcal{E}\}$ denotes the set of immediate predecessors of $v$.

2. **Unlocked Arm Availability.** Each arm $a \in \mathcal{A}$ performs at most one operation at a time. Operation $v$ is schedulable only if all arms in $\mu(v)$ stay free during execution, formally:
$$\forall v, v' \in \mathcal{V}, \ v \neq v', \ \mu(v) \cap \mu(v') \neq \varnothing \ \Rightarrow \ [\sigma(v), \sigma(v) + t_v) \cap [\sigma(v'), \sigma(v') + t_{v'}) = \varnothing. \quad (6)$$

3. **Arm Lock Compatibility.** Each `pick-use-place` sequence involving the same object must be executed by the same arm(s):

$$\forall(v_{\text{pick}}, v_{\text{use}}, v_{\text{place}}) \in \mathbb{S}, \quad \mu(v_{\text{pick}}) = \mu(v_{\text{use}}) = \mu(v_{\text{place}}), \tag{7}$$

where $\mathbb{S}$ denotes the set of all `pick-use-place` triples extracted from $\mathcal{G}$.

4. **Deadlock Prevention.** If a dual-arm task $v_d$ becomes ready but cannot be executed because each arm $a \in \mu(v_d) = \{L, R\}$ is currently locked to a different object, a potential deadlock is detected. To resolve it, we identify the `pick` action $v_{\text{late}}$ that was executed later among the two conflicting chains and rolls it back along with its dependent subtree:

$$\text{Rollback}(v_{\text{late}}) := \text{Descendants}(v_{\text{late}}) \cup \{v_{\text{late}}\}. \tag{8}$$

These removed operations are then re-inserted into the scheduling queue $\mathcal{Q}$, freeing the related arm and allowing the earlier pick chain to proceed. The dual-arm task $v_d$ is subsequently rescheduled once both arms are synchronized and one side hold the object chain.

**Execution Procedure.** Scheduling proceeds iteratively by selecting feasible task pairs from $\mathcal{Q}$ and assigning them to arms based on priority and constraints. For dual-arm tasks $\delta_v = 2$, both arms synchronize: $\sigma_L(v) = \sigma_R(v) = \sigma(v)$. After each pick, object lock status updates, binding all related tasks to the same arm (Constraint 3). If no feasible pair exists due to conflicts or readiness issues, rollback is triggered per Constraint 4, favoring earlier actions. The full scheduling and rollback algorithm is detailed in Appendix Alg. 4.

# 4 CROSS-SCENARIO DUAL-ARM PARALLEL TASK DATASET

To benchmark our new task and method, we introduce the first dataset for long-horizon parallel optimization, covering 10 diverse scenes: kitchen, agricultural greenhouse, office, factory, supermarket, hospital, disaster rescue, hotel, pet shop and library. Each scene presents distinct task planning challenges, which are further exacerbated when handling multi-task instructions. The dataset is structured across three difficulty levels—easy, medium, and hard, where task complexity increases with the number of steps required, allowing for a structured assessment of the method across tasks with varying complexity and scene diversity. The dataset emphasizes efficient parallel operations and prioritizes the effectiveness of dual-arm coordination in multi-task scenarios. Fig. 3 provides a statistical evaluation of our X-DAPT dataset. See Appendix Sec. B for detailed dataset information.

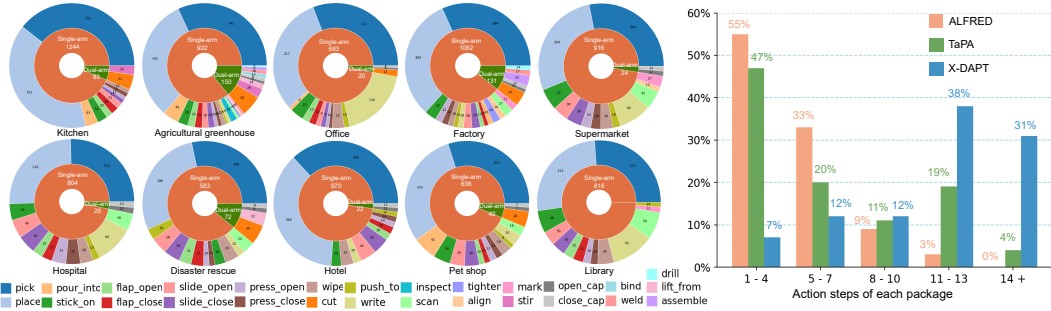

Figure 3: Statistical evaluation of X-DAPT dataset. The pie charts show the appearing skills (outer circle) and required number of arms (inner circle) of the 10 scenes. The bar chart shows the percentage of instructions with different action steps. X-DAPT is larger in scale, richer in scenes, and more diverse in skills than previous datasets (Wu et al., 2023; Shridhar et al., 2020). It is closely aligned with daily life and specifically designed to highlight parallel execution.

Table 1: **Quantitative results of dual-arm task planning under 4 scenes with GPT-4o foundation model.** ↑: higher is better, ↓: lower is better. The red , orange , and yellow colors denote the best, the second best, and the third best results, respectively.

| Scenes | Kitchen | | | | Office | | | | Agricultural Greenhouse | | | | Factory | | | |
|---|---|---|---|---|---|---|---|---|---|---|---|---|---|---|---|---|
| | TEI ↑ | TFR ↓ | PPR ↑ | APR ↑ | TEI ↑ | TFR ↓ | PPR ↑ | APR ↑ | TEI ↑ | TFR ↓ | PPR ↑ | APR ↑ | TEI ↑ | TFR ↓ | PPR ↑ | APR ↑ |
| LLM[3] | 0.805 | 0.333 | 0.000 | 0.069 | 0.985 | 0.267 | 0.000 | 0.009 | 0.785 | 0.400 | 0.000 | 0.026 | 0.913 | 0.012 | 0.000 | 0.000 |
| ChatGPT-Prompts | 0.817 | 0.081 | 0.000 | 0.010 | 1.024 | 0.160 | 0.000 | 0.013 | 0.934 | 0.112 | 0.000 | 0.032 | 0.789 | 0.190 | 0.000 | 0.013 |
| VOYAGER | 0.000 | 1.000 | 0.000 | 0.000 | 0.000 | 1.000 | 0.000 | 0.000 | 0.000 | 1.000 | 0.000 | 0.000 | 0.000 | 1.000 | 0.000 | 0.000 |
| Embodied TaPA | 0.859 | 0.200 | 0.000 | 0.080 | 0.929 | 0.333 | 0.000 | 0.019 | 0.923 | 0.073 | 0.000 | 0.029 | 0.927 | 0.004 | 0.000 | 0.013 |
| LLM-Planner | 0.858 | 0.200 | 0.000 | 0.077 | 1.077 | 0.000 | 0.000 | 0.019 | 0.875 | 0.152 | 0.000 | 0.027 | 0.991 | 0.214 | 0.000 | 0.116 |
| FLTRNN | 0.912 | 0.000 | 0.000 | 0.096 | 1.014 | 0.200 | 0.000 | 0.019 | 0.934 | 0.009 | 0.000 | 0.032 | 0.915 | 0.005 | 0.000 | 0.013 |
| RoCo | 0.709 | 0.153 | 0.000 | 0.016 | 0.979 | 0.173 | 0.000 | 0.010 | 0.890 | 0.142 | 0.000 | 0.058 | 0.845 | 0.148 | 0.000 | 0.013 |
| **RoboPARA** | 1.407 | 0.000 | 0.211 | 0.334 | 1.553 | 0.114 | 0.070 | 0.293 | 1.217 | 0.001 | 0.000 | 0.365 | 1.386 | 0.057 | 0.000 | 0.360 |

# 5 EXPERIMENTS

## 5.1 EXPERIMENTAL SETUP

We utilize the m3e-base[1] model as the memory retriever, and employ OpenAI GPT-4o[2] and DeepSeek V3[3] APIs as foundation LLMs. For the System-1 phase, we employ the ACT (Zhao et al., 2023a), DP (Zhao et al., 2023b), and $\pi_0$ (Black et al., 2024) algorithms to train the action model. To evaluate RoboPARA, we conduct tests across four scenes and perform hardware verification using a humanoid robot, Franka Research 3 (GmbH, 2024), and UR5e (Robots, 2024) robotic arms (see Appendix Sec. E.1 for details).

## 5.2 BASELINES

Seven representative LLM-based agents are adopted as baseline algorithms. We integrated the task settings and constraints into the prompts of all baseline methods with minimal necessary modifications, while retaining their core methodologies, to evaluate the robot's long-horizon planning capability. A detailed prompt comparison between RoboPARA and prior works can be found in Appendix Table 4.

LLM[3] (Wang et al., 2024) uses LLM to jointly generate and iteratively refine symbolic actions and motion parameters based on failure feedback. We provide it with our environment states and fit our tasks and scenes to its background settings. ChatGPT-Prompts (Wake et al., 2023) iteratively translate human instructions into structured robot actions using ChatGPT. We provide the prompt with our execution constraints and overall settings. VOYAGER (Wang et al., 2023) is a GPT-4-powered lifelong agent in Minecraft that explores reusable skills through iterative self-refinement. We re-implement VOYAGER to do task decomposition under our experimental scenes. Embodied TaPA (Wu et al., 2023) uses multi-view perception and GPT-3.5 to generate grounded action plans in physical scenes. We adapt it with GPT-4o and DeepSeek V3 for task decomposition. LLM-Planner (Song et al., 2023) generates subgoals with GPT-3 and replans upon failures. We re-implement it using GPT-4o and DeepSeek V3 for our scenarios. FLTRNN (Zhang et al., 2024) uses RNN framework to enhance long-horizon planning through subgoal decomposition, rule tracing, and external memory. We simply replace the environment state and tasks. RoCo (Mandi et al., 2023) lets agents communicate via dialogue to decompose tasks, allocate goals, and iteratively refine plans through replanning. We map the two agents, *Chad* and *Dave*, to our two robotic arms.

## 5.3 EVALUATION RESULTS

We systematically evaluate RoboPARA and baselines on their planning efficiency, failure rate, parallelism manipulating across packages, and dual-arm simultaneous execution. Formal definitions of the metrics are explained in Appendix Sec. E.1.1. We present the quantitative results with GPT-4o foundation under different scenes and package difficulties in Table 1 and Table 2, respectively. Results with DeepSeek V3 foundation is presented in Appendix Sec. E.4. We proceed with a closer analysis.

---

[1] https://huggingface.co/moka-ai/m3e-base

[2] https://openai.com/index/hello-gpt-4o/

[3] https://huggingface.co/deepseek-ai/DeepSeek-V3

Table 2: **Quantitative results of dual-arm task planning under different task difficulties with GPT-4o foundation model.** ↑: higher is better, ↓: lower is better. The red , orange , and yellow colors denote the best, the second best, and the third best results, respectively.

| Package Difficulty | Easy Packages | | | | Medium Packages | | | | Hard Packages | | | |
|---|---|---|---|---|---|---|---|---|---|---|---|---|
| | TEI ↑ | TFR ↓ | PPR ↑ | APR ↑ | TEI ↑ | TFR ↓ | PPR ↑ | APR ↑ | TEI ↑ | TFR ↓ | PPR ↑ | APR ↑ |
| LLM[3] | 1.406 | 0.000 | 0.000 | 0.051 | 0.829 | 0.300 | 0.000 | 0.014 | 0.382 | 0.459 | 0.000 | 0.012 |
| ChatGPT-Prompts | 1.349 | 0.035 | 0.000 | 0.020 | 0.869 | 0.123 | 0.000 | 0.011 | 0.455 | 0.249 | 0.000 | 0.020 |
| VOYAGER | 0.000 | 1.000 | 0.000 | 0.000 | 0.000 | 1.000 | 0.000 | 0.000 | 0.000 | 1.000 | 0.000 | 0.000 |
| Embodied TaPA | 1.367 | 0.050 | 0.000 | 0.051 | 0.910 | 0.150 | 0.000 | 0.024 | 0.451 | 0.258 | 0.000 | 0.031 |
| LLM-Planner | 1.494 | 0.000 | 0.000 | 0.119 | 0.952 | 0.013 | 0.000 | 0.032 | 0.404 | 0.412 | 0.000 | 0.028 |
| FLTRNN | 1.406 | 0.000 | 0.000 | 0.051 | 0.943 | 0.050 | 0.000 | 0.029 | 0.481 | 0.111 | 0.000 | 0.039 |
| RoCo | 1.226 | 0.110 | 0.000 | 0.029 | 0.888 | 0.131 | 0.000 | 0.020 | 0.454 | 0.221 | 0.000 | 0.024 |
| **RoboPARA** | 1.956 | 0.025 | 0.057 | 0.342 | 1.347 | 0.063 | 0.033 | 0.321 | 0.869 | 0.041 | 0.121 | 0.351 |

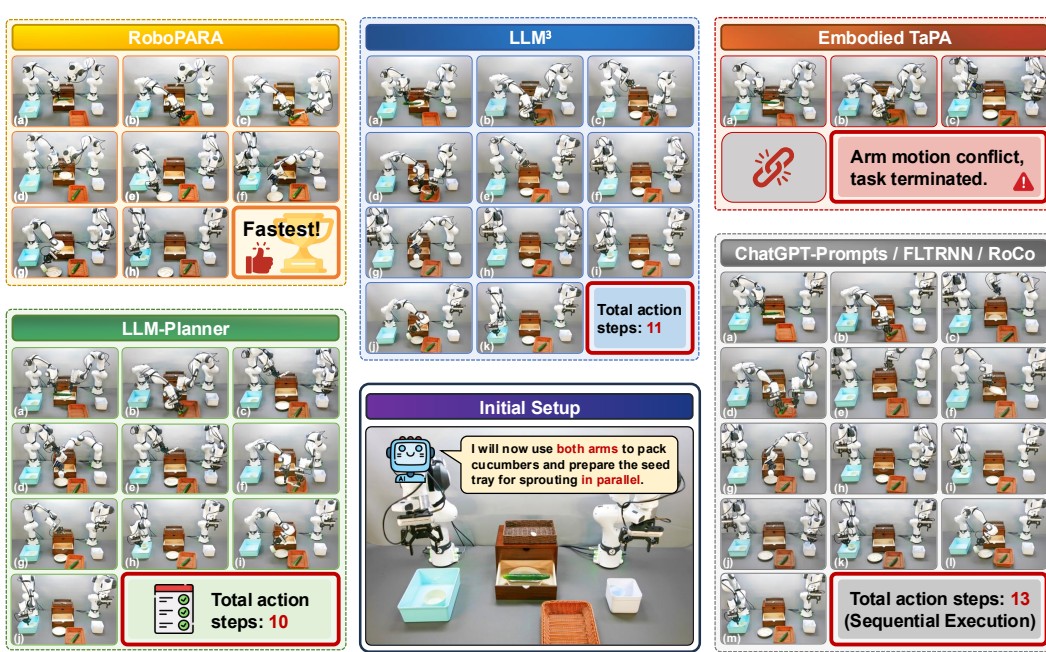

Figure 4: Real-world deployment comparison of RoboPARA and baselines on Franka Research 3 robot in an agricultural greenhouse scene. Voyager is excluded because it cannot provide explicit plans for each arm. See Appendix Sec. E.7.2 for details and specific plans regarding this field test.

**Significantly higher efficiency.** RoboPARA's superiority is evident in its ability to achieve substantial reduction in execution time while maintaining a high task success rate (Fig. 1). RoboPARA consistently achieves at least 1.5×, 1.4×, 1.3×, and 1.4× the number of successful steps executed per unit time compared to baseline methods in the kitchen, office, greenhouse, and factory scenes, respectively. On the other hand, methods such as FLTRNN and ChatGPT-Prompts increase their failure rates as they strive to improve execution efficiency on harder cases.

**Robust failure resilience.** The DAG output verification algorithm (Appendix Alg. 2) plays a crucial role in timely detecting node dependency and resource usage conflicts, ensuring the feasibility and accuracy of task planning. Especially in hard packages, baseline methods exhibit planning failure rates over 2.7× higher than RoboPARA, highlighting the lack of robustness in LLM-only agent approaches when faced with complex planning scenarios. VOYAGER fails to generate concrete execution steps, while RoCo is almost unable to solve any task after more than 50 prompting iterations.

**Integrated execution across packages.** RoboPARA effectively reasons over both intra-package logic and inter-package dependencies in tasks, enabling successful merging of redundant operations. This reduces unnecessary repetition, in contrast to baseline methods that treat each package as an isolated unit (e.g., two tasks with `pick knife → cut carrots → place knife` and `pick knife → cut apples → place knife`, can be merged into a single optimized sequence:

Table 3: Ablation study of our method in kitchen scene. *Upper block*: Ablation studies for environment information and instructional constraints. RoboPARA outperforms almost all the alternatives, demonstrating the critical role of each prompt component. *Lower block*: Ablation studies for the arm selector logic and scheduling algorithm. RoboPARA surpasses almost all the other options, showing the significance of iterative LLM interaction and deadlock-aware algorithmic design.

| Package difficulty | Easy Packages | | | | Medium Packages | | | | Hard Packages | | | |
|---|---|---|---|---|---|---|---|---|---|---|---|---|
| | TEI ↑ | TFR ↓ | PPR ↑ | APR ↑ | TEI ↑ | TFR ↓ | PPR ↑ | APR ↑ | TEI ↑ | TFR ↓ | PPR ↑ | APR ↑ |
| w/o env info | 0.535 | 0.080 | 0.181 | 0.281 | 0.363 | 0.182 | 0.035 | 0.373 | 0.426 | 0.588 | 0.381 | 0.151 |
| w/o instructional constraints | 1.379 | 0.123 | 0.000 | 0.272 | 0.467 | 0.742 | 0.000 | 0.086 | 0.316 | 0.676 | 0.007 | 0.041 |
| w/o deadlock check | 0.855 | 0.447 | 0.170 | 0.234 | 1.061 | 0.184 | 0.056 | 0.353 | 0.428 | 0.618 | 0.402 | 0.156 |
| w/o DAG correction | 1.772 | 0.000 | 0.181 | 0.315 | 1.345 | 0.083 | 0.038 | 0.474 | 0.612 | 0.488 | 0.438 | 0.177 |
| full model | 1.785 | 0.000 | 0.181 | 0.321 | 1.469 | 0.000 | 0.038 | 0.457 | 0.965 | 0.000 | 0.412 | 0.225 |

pick knife → cut carrots → cut apples → place knife). In our experiments, RoboPARA reduces the number of executed steps by 7% while ensuring correct task completion.

**Dual-arm simultaneous execution.** RoboPARA demonstrates superior dual-arm parallelism across all scenes. In simple packages, its number of parallel steps is over $2.8\times$ that of all baselines; in medium and complex packages, this advantage increases to over $10\times$. Baseline methods tend to fall back to sequential execution when processing complex package prompts, due to their inability to model step dependencies. While Embodied TaPA and LLM-Planner attempt to increase parallel execution, they suffer from elevated failure rates, generating infeasible plans. Under the DeepSeek V3 backbone (Appendix Table 8 and Appendix Table 9), baseline parallelism improves significantly, yet RoboPARA still outperforms all of them by over $1.3\times$. We present the comparison between RoboPARA and baselines on real Franka Research 3 robotic arms in Fig. 4.

**Robustness across scenes, difficulty levels, and model backbones.** RoboPARA consistently outperforms all baselines across evaluation metrics under both GPT-4o and DeepSeek V3 backbones (Appendix Fig. 7 and Fig. 6), demonstrating strong robustness.

**Others.** Due to space limitations, additional results are provided in Appendix E.5. These include the full set of experiments on the X-DAPT dataset, evaluations on the RoboTwin (Mu et al., 2025) dataset, as well as extended real-world trials across a broader range of scenarios. Additionally, we report results on closed-loop error handling and replanning, demonstrating RoboPARA's ability to adapt to execution failures and dynamically adjust task plans.

## 5.4 ABLATION STUDIES

We ablate two prompt choices (environment information and instructional constraints) and two algorithm components (arm selector deadlock check and DAG correction) in RoboPARA and study their impact on planning performance (see Appendix Sec. E.3 for details of each ablated variant). Results can be found in Table 3. We highlight the key findings below:

- **Environment information is crucial in ensuring correction and enhancing parallelism.** Failure rate increases by 28% and parallel execution decreases by 25% without environment information in prompts. This finding corroborates recent studies in the literature (Mandi et al., 2023).

- **Instructional constraints and DAG correction algorithm ensures successful plannings.** Removing either of the two mechanisms individually leads to a significant increase (on average 51% and 18%) in failure rates, especially on hard packages. Prompt constraints and the verification algorithm are critical for generating correct DAG nodes, as incorrect node dependencies can result in deadlocks or infinite loops during the scheduling process.

- **Arm selector (Appendix Alg. 3) is the core component of RoboPARA framework.** Without timely deadlock detection and case-specific reasoning, execution efficiency drops by an average of 22%, while the failure rate increases by an average of 18%.

## 5.5 LIMITATION AND FUTURE WORK

The limitations of RoboPARA primarily lie in cost, accuracy, and generalization. First, the use of GPT-4o and DeepSeek V3 APIs incurs significant costs, with RoboPARA consuming an average

of $1.3\times$ more tokens than baselines due to iterative DAG corrections. Second, despite the iterative prompting mechanism, inaccuracies persist as the agent struggles to self-correct planning errors for complex packages with intricate dependencies within the limited correction rounds. Finally, the current skill library is tightly coupled to predefined scenarios and package templates, limiting its ability to generalize to unseen tasks or environments. Looking ahead, advancements in LLM, along with more adaptive DAG generation and verification mechanisms, hold promise for addressing these challenges and enabling RoboPARA to scale across a wider range of dual-arm robotic applications.

## 6  CONCLUSION

In this work, we propose RoboPARA, a novel LLM-driven framework to address the Dual-Arm Cooperative Scheduling Problem. RoboPARA employs a two-stage process: (1) Dependency Graph-based Planning, which models task dependencies and eliminates redundancy, and (2) Graph Re-Traversal-based Parallel Planning, which maximizes parallelism while maintaining task coherence. To evaluate its effectiveness, we introduce the X-DAPT dataset, designed to benchmark dual-arm task parallelism across diverse scenarios. Experimental results show that RoboPARA significantly outperforms existing methods in efficiency and reliability, particularly for complex tasks, demonstrating its potential for real-world dual-arm robotic applications.

## ACKNOWLEDGEMENTS

This work was supported by the New Generation Artificial Intelligence–National Science and Technology Major Project (Grant No. 2025ZD0122603). It was also supported by the Postdoctoral Fellowship Program and the China Postdoctoral Science Foundation (Grant Nos. 2024M764093 and BX20250485), the Beijing Natural Science Foundation (Grant No. 4254100), and the Beijing Advanced Innovation Center for Future Blockchain and Privacy Computing. This work was also supported by Beijing Innovation Center of Humanoid Robotics.

## REPRODUCIBILITY STATEMENT

We have taken extensive measures to ensure the reproducibility of our work. Section E.6 outlines the experimental setup (including the knowledge base construction across 4 initial scenes, task package grouping by difficulty, and test protocol design), adopted backbone models (m3e-base as the RAG retriever, GPT-4o and DeepSeek V3 as foundation LLMs), hardware environment (Franka Research 3, UR5e robotic arms, and humanoid robot), and baseline configurations (7 representative LLM-based agents with adapted prompts), with additional implementation details (such as the full prompts for RoboPARA and baselines) and adopted hyperparameters (*e.g.*, retry limits for DAG correction in Stage 1) provided in Appendix A and Appendix E. All datasets used in our study, including the newly proposed Cross-Scenario Dual-Arm Parallel Task dataset (X-DAPT dataset) and the reorganized RoboTwin dataset, are detailed in Section 4, Appendix B and Appendix E.5.2, with X-DAPT spanning 10 diverse scenarios and 3 difficulty levels. To further facilitate reproducibility, we include the complete source code (for DAG generation, graph re-traversal scheduling, and closed-loop error handling), all experiment scripts (for scene-specific and difficulty-specific tests), and relevant model configurations in the SUPPLEMENTARY MATERIALS, and we will make the full codebase and resources publicly available upon publication.

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

# A   METHODOLOGY

## A.1   ROBOPARA ALGORITHM

The RoboPARA framework comprises two stages: **Dependency Graph-based Planning Candidates Generation** and **Graph Re-Traversal-based Dual-Arm Parallel Planning**. The overall pseudocode is presented in Alg. 1.

---

**Algorithm 1** RoboPARA Algorithm

---

**Input:** Natural language instruction $\mathcal{I}$
$\qquad\qquad\qquad\qquad\qquad\qquad$ ▷ **Stage 1: Dependency Graph-based Planning Candidates Generation**
$\mathcal{S} \leftarrow \text{SCENEIDENTIFY}(\mathcal{I})$ $\qquad\qquad\qquad\qquad\qquad\qquad\qquad$ ▷ Extract task-related scene
$\mathcal{R} \leftarrow \text{RAGGENERATE}(\mathcal{I}, \mathcal{S})$ $\qquad\qquad\qquad\qquad$ ▷ Retrieve and generate scene-specific description
$\mathcal{E} \leftarrow \text{ENVDESCRIPTION}(\mathcal{S})$ $\qquad\qquad\qquad\qquad\qquad$ ▷ Load environment constraints
$\qquad\qquad\qquad\qquad\qquad\qquad\qquad\qquad$ ▷ DAG Construction and Correction
$\mathcal{P}_0 \leftarrow \text{PROMPTFORMAT}(\mathcal{R}, \mathcal{E})$ $\qquad\qquad\qquad\qquad\qquad$ ▷ Formulate first prompt
$\mathcal{G}_0 \leftarrow \text{STRUCTUREDGENERATE}(\mathcal{P}_0)$ $\qquad\qquad\qquad$ ▷ Obtain initial DAG description
$\mathcal{N}_0 \leftarrow \text{BUILDGRAPH}(\mathcal{G}_0)$ $\qquad\qquad\qquad\qquad\qquad$ ▷ Parse DAG nodes
$r \leftarrow \text{VERIFYGRAPH}(\mathcal{N}_0)$ $\qquad\qquad\qquad\qquad\qquad$ ▷ Check structural consistency

**while** $r \neq 0$ and $c_{\text{retry}} \leq 2$ **do**
$\qquad \mathcal{P}_{\text{corr}} \leftarrow \text{GENERATECORRECTIONPROMPT}(\mathcal{N}_0, r)$
$\qquad \mathcal{G}_{\text{corr}} \leftarrow \text{STRUCTUREDGENERATE}(\mathcal{P}_{\text{corr}})$
$\qquad \mathcal{N}_0 \leftarrow \text{BUILDGRAPH}(\mathcal{G}_{\text{corr}})$
$\qquad r \leftarrow \text{VERIFYGRAPH}(\mathcal{N}_0)$
$\qquad c_{\text{retry}} \leftarrow c_{\text{retry}} + 1$
**end while**
$\qquad\qquad\qquad\qquad\qquad$ ▷ **Stage 2: Graph Re-Traversal-based Dual-Arm Parallel Planning**
$(\text{Plan}_{\text{left}}, \text{Plan}_{\text{right}}, \mathcal{T}_{\text{total}}) \leftarrow \text{SCHEDULE}(\mathcal{N}_0)$
**Output:** $\text{Plan}_{\text{left}}, \text{Plan}_{\text{right}}, \mathcal{T}_{\text{total}}$

---

## A.2   STAGE 1: DEPENDENCY GRAPH-BASED PLANNING CANDIDATES GENERATION

OpenAI ChatGPT-4o allow users to specify the role of each prompt message as one of three types:

- System: High-level instructions that define the model's behavior and tone across the entire interaction.

- User: Specific instructions that guide the assistant's next response.

- Assistant: The model-generated reply. For more details, see the OpenAI Chat Completions Guide.

To reduce token consumption, we avoid multi-turn interactions by concatenating the system and user prompts into a single input when generating each assistant response. Check `https://platform.openai.com/docs/guides/chat/introduction` for detailed information.

### A.2.1   COMPONENTS IN THE PROMPT

We design two prompt templates to guide the LLMs in **Stage 1: Dependency Graph-based Planning Candidates Generation**. Below we detail the key components of each template.

**Template 1: Structured Package Step Generation**. In this template, a structured prompt is used to convert natural language instructions into step-wise package descriptions. The prompt is carefully designed to include:

- **Scene Retrieval:** The LLM retrieves the corresponding environment context (e.g., `kitchen`, `office`) based on the instruction.

- **Example-based Alignment:** An in-context example is given to help the LLM generate consistently structured outputs (step ID, action, source/target, arm number, time).

- **Error Resilience:** The LLM is explicitly prompted to revise any logical or ordering mistakes in the given instruction.
- **Canonical Output Format:** The response is constrained to a rigid textual format (e.g., `A1: pick(source=..., target=...)(Single arm, 5 seconds))`, enabling downstream parsing.

This template transforms vague instructions into semi-structured procedural steps, serving as the intermediate grounding between human intent and robotic execution. See Sec. A.2.2 for the full prompt.

**Template 2: Dependency-Aware Graph Construction and Refinement**. Based on the generated package steps and retrieved environment state, a second prompt is issued to the LLM to construct a valid DAG that captures execution dependencies. The prompt incorporates the following elements:

- **Environment-Aware Reasoning:** All assets and object locations are embedded into the prompt to ground spatial and contextual preconditions.
- **Pick-Use-Place Semantics:** Nodes are required to follow a strict `pick-use-place` chain. Tool usage must follow prior pick actions, and multiple uses of the same tool must form a linear dependency.
- **Edge Constraints:** `pick` nodes can have multiple edges; `use` and `place` nodes must have exactly one dependency.
- **Tool Sharing Optimization:** Shared tools (e.g., `knives`) should be reused across compatible steps to reduce motion cost.
- **Container Efficiency:** The model is instructed to minimize redundant open or close operations by grouping relevant actions together.
- **Temporal Encoding:** Waiting steps are encoded as `delay_after` fields in the dependent node, avoiding redundant nodes.
- **Multi-Round Correction:** If the initial DAG violates structural rules (e.g., tool usage relying on unrelated objects), error messages are auto-generated and used to prompt a corrected version.

This template transforms procedural steps into DAG nodes with awareness of execution dependencies. See Sec. A.2.2 for the full prompt.

Overall, our prompting framework transforms high-level task goals into executable plans with explicit structure and robot-aware constraints. We conduct a prompt comparison in Table 4 with prior works.

Table 4: **Prompt component comparison between RoboPARA and prior works.** Embodied TaPA is not included because its prompt does not include any listed components.

| Prompt Components | LLM[3] | ChatGPT-Prompts | VOYAGER | LLM-Planner | FLTRNN | RoCo | **RoboPARA** |
|---|---|---|---|---|---|---|---|
| Environment Information | ✓ | ✓ | | | ✓ | ✓ | ✓ |
| Instructional Constraints | | ✓ | | ✓ | ✓ | | ✓ |
| Skill Library | ✓ (pre-defined) | ✓ (pre-defined) | | ✓ (pre-defined) | ✓ (pre-defined) | ✓ (pre-defined) | ✓ (RAG generated) |
| Output Format | | | ✓ (JSON) | | | | ✓ (DAG) |
| Memory Management | | ✓ | | ✓ | | | ✓ |
| Examples | | | ✓ | | ✓ | | ✓ |
| Visible Objects | | | | ✓ | | | |

### A.2.2 FULL PROMPT

> **Template 1: Structured Package Step Generation.**
>
> ```
> You need to adjust the package steps: {context}
> The order of the packages and the steps within the packages that I
> provided may contain some issues. Please revise them accordingly and
> output the corrected result.
> Example format:
> factory scene
> ```

```
Package A: Assemble Tools
A1:  pick(source="tool_rack", target="wrench")(Single arm, 5 seconds)
A2:  place(source="wrench", target="assembly_table")(Single arm, 6
seconds)
A3:  pick(source="tool_rack", target="screwdriver")(Single arm, 5
seconds)
A4:  place(source="screwdriver", target="assembly_table")(Single arm,
6 seconds)
factory scene
Package B: Prepare Tools
B1:  pick(source="tool_shelf", target="voltmeter")(Single arm, 5
seconds)
B2:  place(source="voltmeter", target="workbench")(Single arm, 6
seconds)
B3:  pick(source="drawer", target="cable")(Single arm, 5 seconds)
B4:  place(source="cable", target="tool_holder")(Single arm, 5
seconds)
factory scene
Package C: Wipe Surface
C1:  pick(source="cleaning_station", target="rag")(Single arm, 5
seconds)
C2:  wipe(source="rag", target="assembly_table")(Single arm, 10
seconds)
C3:  place(source="rag", target="cleaning_station")(Single arm, 5
seconds)
Do not output any extra text--only the packages with the corrected
step order.
```

---

### Template 2a: Dependency-Aware Graph Construction (First Call).

```
You are a right-hand man for a two-armed robot to combine several
well-defined packages into a directed acyclic graph (DAG).
You need to convert all the package steps I entered into you into a
DAG chart.
```
**Environment Information:** {target_env}
**Missions:** {instruction}{rag_output}
```
You must follow the following criteria:
```

1) Create a DAG according to the package steps, and consider all
   nodes to complete the task.

2) A tool usage node can depend on another tool usage node that uses
   the same source object, or on a "pick" node whose target object
   is the same as the tool usage node's source object. When there
   are multiple tool usage nodes, the first one should depend on the
   "pick" node, while each subsequent one depends on the previous
   tool usage node. A "place" node can depend on a "pick" node whose
   target matches the "place" node's source, or on another "place"
   node that shares the same source object. For example, when adding
   water to a pot, the pot should be placed first, then the water
   picked up. Therefore, the water's "pick" node should depend on
   the pot's "place" node, rather than the water's "pour_into" node
   depending on a tool usage node involving the pot.

3) Each robotic arm can only hold one object at a time and must place
   it down before picking up another object. After picking up an
   object, the next step must be either to use it or to place it.
   You do not need to strictly follow the step-by-step order of the
   package; you may arrange steps flexibly as long as the dependency
   rules outlined above are followed.

4) A "pick" node can have multiple dependency edges, but a tool
   usage node and a "place" node can each have only one dependency
   edge. In other words, the "edge" field of a pick node can include
   multiple nodes, whereas for tool usage and place nodes, the edge
   field must contain only one node.

5) Tool usage nodes include actions such as push_to, wipe, stick_on, pour_into, cut, stir, etc. Please refer to the skill set below for a full list.

6) Nodes are divided into four types, container switch node, tool use node, take node, place node. The nodes used by the tool include push_to, wipe, stick_on, pour_into, cut, stir, etc. The operations for opening and closing containers are slide_open, slide_close, flap_open, flap_close, open_cap, close_cap, etc. The pick node is "pick" and the place node is "place". Dependencies need to follow a pick-use-place form.

7) The tool use node only relies on the "pick" node of the tool when it is used only once. When the tool is used multiple times, the first tool uses the dependency tool "pick" operation, and the subsequent tool use node depends on the last tool use node. The "place" node can only depend on the use of the object or the "pick" node.

8) The generated edge is a prerequisite not only for executing the node, but also for completing the task objective. For example, in the task of cutting apples and cucumbers, the apple and banana are placed on the cutting board before taking the knife and cutting.

9) The steps using the same tool in different packages can be combined under certain conditions, such as using a knife to cut the object to be cut, using a brush to brush the object at one time, which can save the overall execution time.

10) The cutting operation can only begin when all objects are placed in the designated area, and all objects need to be ready when making the first cut. For example, if you want to cut mushrooms, apples, cucumbers, you need to put the mushrooms, apples, and cucumbers on the cutting table during the first cut and then start cutting. The same is true for wiping.

11) Minimize container interaction: After opening the container, retrieve all the objects you need, and then close it. When putting items back into the container, make sure all items are placed before closing. Avoid redundant on/off cycles.

12) The direct dependent chain of the picked up object: After the picked up object, all subsequent nodes must be directly involved in its use or placement. Do not insert unrelated steps (such as closing the door, handling other objects) between picking up and putting down.

13) "Take" and "put" are always interdependent, because you can't put something down until you pick it up. Pick up an object, use the object, and then drop the object.

14) Do not create nodes when waiting operations (such as waiting for baking) are encountered. Instead, the 'delay_after' property is added to subsequent nodes that depend on the wait period. For example, if step C12 is the wait after C11 (turning on the oven) and before C13 (turning off the oven), the wait is merged into C13 by setting delay_after=600 and edge=[C11] for C13.

15) The operations of each object must form a continuous Pick-Use-Place chain. When the Use operation requires a tool (such as cutting with a knife), for example, when cutting carrots, the cut node should rely on the pick of the knife, and the pick operation depends on the place of the carrots.

16) The Use node should rely on the Pick step of the "source" object, but not on the Pick and Place steps of the "target" object, and the fetch and place operation of the "target" object of the Use operation should be completed before the fetch of the "source" object. For example stick_on(source="cheese", target="bread"). place(source="bread", target="plate") should be completed before pick(source="refrigerator", target="cheese").

```
17) Use tool nodes include: push_to, wipe, stick_on, pour_into, cut,
    stir, etc.  You can't use a tool until you pick it.

18) Once you've put everything on the cutting board, take the knife
    and try to pick it up only once, cutting everything you need to
    cut before putting the knife down.

19) After the graph is generated, create a task completion node.
```

**List of skills:** {List of skills}

```
You should only reply in the following format and do not reply to
anything else.
Do not use json format output.
```

**Response format:**

```
Nodes:
node_index = 1, 2, 3...    :
type:  indicates the node type.
name:  indicates the node name.
arm_num:  specifies the arm number of a node.
take_time:The time required for each step.
edge:  Edge list of nodes.

Example:
Missions:  Complete Package A.
Package A: Make carrot slices
A1:  pick(source="table", target="carrots")(Single arm, 5 seconds)
A2:  place(source="carrots", target="cutting_board")(Single arm, 7
seconds)
A3:  pick(source="counter", target="knife")(Single arm, 5 seconds)
A4:  cut(source="knife", target="carrots")(Dual arm, 10 seconds)
A5:  place(source="knife", target="counter")(Single arm, 5 seconds)
Example output:
Nodes:
node_1:
type:  pick
name:  pick(source="table", target="carrots")
arm_num:  1
take_time:  5
edge:  []

node_2:
type:  place
name:  place(source="carrots", target="cutting_board")
arm_num:  1
take_time:  7
edge:  [1]

node_3:
type:  pick
name:  pick(source="counter", target="knife")
arm_num:  1
take_time:  5
edge:  [2]

node_4:
type:  cut
name:  cut(source="knife", target="carrots")
arm_num:  2
take_time:  10
edge:  [3]

node_5:
type:  place
name:  place(source="knife", target="counter")
arm_num:  1
take_time:  5
```

```
edge:  [4]

node_6:
type:  task_completion
name:  task_completion
arm_num:  0
take_time:  0
edge:  [5]
```

**Template 2b: Dependency-Aware Graph Construction (Refinements).**

```
{response}
The above is the result of a dual-arm robot decomposing a task into
a DAG graph, where the edge field indicates which other node a
given node depends on.  Some of the nodes in the above DAG contain
dependency errors--in other words, there are problems with the edge
field of certain nodes.  The following are the nodes with issues and
the corresponding problems identified.
{problems_section}
You need to ensure that the DAG graph meets the following
requirements:

1)  All operations between the "pick" and "place" nodes must be
    related to the same object.  No other operations can be performed
    on the object before it has been placed.

2)  Tool usage nodes include:  push_to, wipe, stick_on, pour_into, cut,
    stir, etc.  Container open/close operations include:  slide_open,
    slide_close, flap_open, flap_close, open_cap, close_cap,
    press_open, press_close.  The pickup node is "pick", and the
    placement node is "place".  The dependency relationship must
    follow the pick - use - place pattern.

3)  Tool usage nodes must not depend on the "place" node of another
    object.  If they currently do, move that "place" node into the
    "edge" of the corresponding "pick" node for the tool.

4)  If there is only one tool usage node, it should depend only on the
    tool's "pick" node.  If there are multiple tool usage nodes for
    the same tool, the first tool usage depends on the "pick" node,
    and each subsequent tool usage node depends on the previous tool
    usage node.

5)  The "pick" node for a tool can depend on another object's "place"
    node.

6)  When placing an object back into a container, the container must
    be opened before picking up the object that is to be returned.

7)  A "pick" node can have multiple dependency edges.  Tool usage
    nodes and "place" nodes must each have only one dependency edge.
    In other words:  the "edge" field of a "pick" node can contain
    multiple nodes, but the "edge" field of a tool usage or "place"
    node can contain only one node.

You should only reply in the following format and do not reply to
anything else.
Do not use json format output.
You must output the entire DAG, not just the modified nodes.
Your output format must be:
Nodes:
node_index = 1, 2, 3...    :
type:  indicates the node type.
name:  indicates the node name.
arm_num:  specifies the arm number of a node.
take_time:The time required for each step.
edge:  Edge list of nodes.
Example:  ...(same as first call)
```

### A.2.3 SKILL LIBRARY

The set of skills covered by all task packages retrieved via RAG across our 4 test scenarios is summarized in Table 5.

Table 5: Skill library covering all task packages retrieved via RAG across our test scenarios.

| Skill Name | Parameters | Description | Arm Req. |
|---|---|---|---|
| pick | source, target | Pick the "target" object from the "source" placement. | Left/Right |
| place | source, target | Place the "source" object into/on the "target" area. | Left/Right |
| slide_open | target | Slide open a prismatic-joint object (e.g., drawer). | Left/Right |
| slide_close | target | Slide close a prismatic-joint object (e.g., drawer). | Left/Right |
| flap_open | target | Rotate open a revolute-joint object (e.g., microwave). | Left/Right |
| flap_close | target | Rotate close a revolute-joint object (e.g., oven). | Left/Right |
| push_to | source, target | Push the "target" object to a specified location. | Left/Right |
| lift_from | source, target | Lift the "source" object from the "target" surface. | Dual |
| open_cap | target | Open the cap of the "target" object. | Dual |
| close_cap | target | Close the cap of the "target" object. | Dual |
| wipe | source, target | Wipe the "target" using the "source" object. | Left/Right |
| stick_on | source, target | Stick the "source" object onto the "target". | Left/Right |
| pour_into | source, target | Pour the "source" substance into the "target" container. | Left/Right |
| cut | source, target | Use the "source" tool to cut the "target". | Dual |
| stir | source, target | Use the "source" tool to stir the "target". | Dual |
| press_open | target | Press to open the "target" object. | Left/Right |
| press_close | target | Press to close the "target" object. | Left/Right |
| write | source, target | Use the "source" to write on the "target". | Left/Right |
| inspect | source, target | Inspect the "target" area using the "source". | Left/Right |
| bind | source, target | Bind the "source" to the "target" support. | Dual |
| weld | source, target | Weld the "target" using the "source" welding tool. | Dual |
| scan | source, target | Scan the "target" object with the "source" scanner. | Left/Right |
| tighten | source, target | Tighten the "target" component using the "source". | Left/Right |
| align | target | Align the "target" component to its correct pose. | Left/Right |
| assemble | source, target | Assemble the "target" part with the "source" tool. | Dual |
| drill | source, target | Drill holes into the "target" with the "source" drill. | Dual |
| mark | source, target | Mark the "target" object using the "source". | Left/Right |

**Note:** Arm requirements specify whether tasks can be done by one (Left/Right) or both arms.

### A.2.4 DAG OUTPUT VERIFICATION

We develop a customized rule-based validation algorithm to systematically verify the structural integrity and action closure of LLM-generated DAGs. Our validator ensures that each place node is properly preceded by a corresponding pick, and that all object manipulations comply with single-possession constraints under dual-arm execution. The criteria and conditions considered in our validation are illustrated in Table 6. The pseudocode of our validation algorithm is shown in Alg. 2.

Table 6: Validation criteria and associated conditions in Alg. 2.

| Problem Index | Problem Description |
|---|---|
| P1 | Depends on another object's place node. |
| P2 | Does not depend on the tool usage node but directly depends on the pick node. |
| P3 | Depends on another object's tool usage node. |

---

**Algorithm 2** DAG Output Verification

---

**Input:** DAG node set $\mathcal{N} = \{n_1, \ldots, n_k\}$ generated by LLMs
**Output:** $r \in \{0, 1\}$ indicating whether logical conflicts are found
                                               ▷ Initialize violation sets
$\mathbf{P1}, \mathbf{P2}, \mathbf{P3} \leftarrow \varnothing$
                                  ▷ Check node-level logical consistency

**for** $n_i \in \mathcal{N}$ **do**
    **if** $n_i.\text{type} \notin \mathcal{A}_{\text{switch}}$ **then**
        **for** $n_j \in \mathcal{P}(n_i)$ **do**               ▷ $\mathcal{P}(n)$: predecessors of $n$
            $p_i \leftarrow (n_i.\text{type} = \texttt{pick})$
            $p_j \leftarrow (n_j.\text{type} = \texttt{pick})$
            $s_i \leftarrow n_i.\text{source}, \quad s_j \leftarrow n_j.\text{source}, \quad t_j \leftarrow n_j.\text{target}$
            $u_j \leftarrow \exists n_u \in \mathcal{S}(n_j) \text{ s.t. } n_u.\text{type} \in \mathcal{A}_{\text{use}} \wedge n_u.\text{source} = t_j$
            **if** $\neg p_i \wedge n_j.\text{type} = \texttt{place} \wedge s_i \neq s_j$ **then**
                $\mathbf{P1} \leftarrow \mathbf{P1} \cup \{n_i\}$        ▷ Invalid: wrong object dependency on place
            **else if** $p_j \wedge u_j \wedge n_i.\text{type} = \texttt{place} \wedge s_i = t_j$ **then**
                $\mathbf{P2} \leftarrow \mathbf{P2} \cup \{n_i\}$          ▷ Invalid: place skips required tool-use
            **else if** $n_j.\text{type} \in \mathcal{A}_{\text{use}} \wedge s_i \neq s_j$ **then**
                $\mathbf{P3} \leftarrow \mathbf{P3} \cup \{n_i\}$          ▷ Invalid: cross-object use dependency
            **end if**
        **end for**
    **end if**
**end for**
                                                 ▷ Return verification result
**if** $\mathbf{P1} \neq \varnothing \vee \mathbf{P2} \neq \varnothing \vee \mathbf{P3} \neq \varnothing$ **then**
    **return** 1                                        ▷ Conflicts detected
**else**
    **return** 0                              ▷ All logical constraints passed
**end if**

---

## A.3    STAGE 2: GRAPH RE-TRAVERSAL-BASED DUAL-ARM PARALLEL PLANNING

We define the dual-arm coordination and scheduling policy through two central routines:

- **Arm Selector:** Determines which robot arm (left, right, or both) should execute the upcoming task node based on current system state.

- **Temporal Scheduler:** Simulates time progression via a priority queue and dispatches ready tasks to appropriate arms following dependencies, lock constraints, and deadlock avoidance.

### A.3.1    ARM SELECTOR

This selector implements a symbolic decision rule to resolve which arm(s) should execute a task. It handles several scenarios:

- **Constraint-aware placement:** For `place` tasks, the arm that originally picked the object is forcibly reused to ensure object consistency.

- **Dual-arm execution:** For dual-arm tasks, we attempt to acquire both arms while maintaining consistency of task chains across arms. It allows partial reuse if only one arm holds the source object.

- **Lock-sensitive fallback:** If one arm is locked on a conflicting chain, the algorithm selects the free arm only if it does not violate dependency or chain constraints.

- **Default heuristic:** In unconstrained settings, the selector defaults to the earliest-available arm (preferring left in ties).

The pseudocode of our arm selector is presented in Alg. 3.

### A.3.2    TEMPORAL SCHEDULER

This algorithm follows the execution timeline:

---

**Algorithm 3** Arm Selector $\mathcal{S}(\tau, F, L)$

---

**Domain:** $\mathcal{A} = \{\mathsf{L}, \mathsf{R}\}$               $\triangleright$ Left/right arm
**Return set:** $\Sigma = \{\mathsf{L}, \mathsf{R}, \mathsf{both}, \bot\}$           $\triangleright$ Decision choices
**Input:**
  • $\tau = (\tau_{\text{type}}, \tau_{\text{start}}, \tau_{\text{source}}, \tau_{\text{target}}, \tau_{\text{arms}})$        $\triangleright$ Task descriptor
  • $F : \mathcal{A} \to \mathbb{R}^+$              $\triangleright$ Free-time function
  • $L : \mathcal{A} \to \{\mathsf{locked} \in \mathbb{B}, \mathsf{chain} \in \mathcal{O} \cup \{\bot\}\}$      $\triangleright$ Lock state
**function** $\mathsf{own}(a, o)$         $\triangleright$ Check whether arm $a$ holds object $o$
  **return** $(L_a.\mathsf{chain} = o)$
**end function**
**function** $\mathsf{lock}(a)$            $\triangleright$ Check lock state of arm $a$
  **return** $L_a.\mathsf{locked}$
**end function**
**function** $\mathsf{is\_dual}(\tau)$
  **return** $(\tau_{\text{arms}} = 2)$
**end function**

**if** $\tau_{\text{type}} = \mathsf{place}$ **then**
  **return** $\arg\max_{a \in \mathcal{A}} \nVdash [\mathsf{own}(a, \tau_{\text{source}}) \wedge F(a) \leq \tau_{\text{start}}]$   $\triangleright$ Find arm holding source and ready
**end if**

**if** $\forall a \in \mathcal{A}, \mathsf{lock}(a)$ **then**         $\triangleright$ Case I: both arms locked
  $o^* \leftarrow \begin{cases} \tau_{\text{target}}, & \text{if } \tau_{\text{type}} = \mathsf{pick} \\ \tau_{\text{source}}, & \text{otherwise} \end{cases}$
  $M(a) := \mathsf{own}(a, o^*)$
  **if** $\mathsf{is\_dual}(\tau)$ **then**
   **if** $M(\mathsf{L}) \wedge M(\mathsf{R})$ **then**
    **return** both
   **else if** $M(\mathsf{L}) \wedge L_{\mathsf{R}}.\mathsf{chain} = \bot$ or $M(\mathsf{R}) \wedge L_{\mathsf{L}}.\mathsf{chain} = \bot$ **then**
    **return** both
   **end if**
  **else**            $\triangleright$ Single-arm task in locked setting
   **if** $M(\mathsf{L}) \wedge \neg M(\mathsf{R}) \wedge F(\mathsf{L}) \leq \tau_{\text{start}}$ **then**
    **return** L
   **else if** $M(\mathsf{R}) \wedge \neg M(\mathsf{L}) \wedge F(\mathsf{R}) \leq \tau_{\text{start}}$ **then**
    **return** R
   **else if** $M(\mathsf{L}) \wedge M(\mathsf{R})$ **then**
    **return** $\arg\min_{a \in \mathcal{A}} F(a)$
   **end if**
  **end if**
  **return** $\bot$
**end if**

**if** $\mathsf{is\_dual}(\tau)$ **then**
  **return** both         $\triangleright$ Case II: free dual-arm operation
**end if**

**if** $\exists! a \in \mathcal{A}, \mathsf{lock}(a)$ **then**        $\triangleright$ Case III: only one locked
  Let $a$ be the locked arm, $b = \mathcal{A} \setminus \{a\}$
  **if** $\tau_{\text{type}} = \mathsf{place} \wedge \mathsf{own}(a, \tau_{\text{source}}) \wedge F(a) \leq \tau_{\text{start}}$ **then**
   **return** $a$
  **else if** $\mathsf{own}(a, \tau_{\text{source}}) \wedge F(a) \leq \tau_{\text{start}}$ **then**
   **return** $a$
  **else if** $F(b) \leq \tau_{\text{start}}$ **then**
   **return** $b$
  **else**
   **return** $\bot$
  **end if**
**end if**

**return** $\arg\min_{a \in \mathcal{A}} F(a)$       $\triangleright$ Case IV: fully free, pick earliest

---

- **Event-Driven Time Simulation:** It uses a min-heap priority queue where each event corresponds to either task readiness (available) or task completion (completed).
- **Dependency-Driven Activation:** Tasks are only pushed into the queue when all predecessor nodes have completed.
- **Lock Tracking & Synchronization:** After a `pick`, the arm is marked as "locked" and associated with the corresponding object; After a `place`, the arm is unlocked, releasing the associated object.
- **Deadlock Resolution Mechanism:** A notable innovation is runtime deadlock detection and recovery in scenarios where: a) Two arms each hold a distinct object via `pick`, and b) A subsequent task requires both arms for a dual-arm action involving only one object. In this case, the scheduler identifies the conflict and rolls back one of the previous `pick` operations (by removing its event and schedule entry) to break the circular wait. It then reassigns the dual-arm task to the arm with the matching chain, ensuring forward progress.
- **Delay Propagation:** Each node may specify a soft constraint modeling post-condition stabilization time. This is added to the successors' earliest start time.

By combining fine-grained chain tracking, resource-aware dispatch, and aggressive yet safe deadlock recovery, this scheduling architecture achieves robust task coordination for dual-arm manipulation in complex, multi-object environments. The pseudocode of our scheduler is presented in Alg. 4.

## A.4 OVERALL EXAMPLE

We present an example in the kitchen scene, utilizing the whole RoboPARA framework. For detailed coding implementations, please refer to our codebase in supplementary materials.

---

**Overall Example: RoboPARA input.**

```
Imagine you are a robot programmed for task planning.  You possess
long-term memory:  <knowledge base> and short-term memory <recent
discoveries>.  Your assigned task is:  please make me carrot slices,
apple salad and cream bread.
```

---

**Overall Example: Memory retrieval output. These are the execution instructions of each task package used for DAG generation in Stage 1: LLM-Driven Dependency Graph for Planning Candidates Generation.**

```
Package A: Make carrot slices
A1:  pick(source="table", target="carrots")(Single arm, 5 seconds)
A2:  place(source="carrots", target="cutting_board")(Single arm, 7
seconds)
A3:  pick(ource="knife", target="carrots")(Dual arm, 10 seconds)
A5:  place(source="counter", target="knife")(Single arm, 5 seconds)
A4:  cut(source="knife", target="counter")(Single arm, 5 seconds)

Package B: Make apple salad
B1:  pick(source="table", target="apples")(Single arm, 5 seconds)
B2:  place(source="apples", target="cutting_board")(Single arm, 7
seconds)
B3:  pick(source="counter", target="knife")(Single arm, 5 seconds)
B4:  cut(source="knife", target="apples")(Dual arm, 10 seconds)
B5:  place(source="knife", target="counter")(Single arm, 5 seconds)

Package C: Make cream bread
C1:  pick(source="shelf", target="plate")(Single arm, 5 seconds)
C2:  place(source="plate", target="table")(Single arm, 6 seconds)
C3:  pick(source="table", target="bread")(Single arm, 5 seconds)
C4:  place(source="bread", target="plate")(Single arm, 8 seconds)
C5:  flap_open(target="refrigerator")(Single arm, 3 seconds)
C6:  pick(source="refrigerator", target="butter")(Single arm, 5
seconds)
C7:  flap_close(target="refrigerator")(Single arm, 3 seconds)
```

---

**Algorithm 4** Temporal Scheduler

---

**Input:** node set $\mathcal{N} = \{n_i\}$, each with $\mathsf{type}_i$, $\mathsf{dep}_i$, $\mathsf{succ}_i$, $\mathsf{take}_i$, $\mathsf{start}_i$, $\mathsf{arms}_i$, $\mathsf{source}_i$, $\mathsf{target}_i$
**Initialize:**
$\mathcal{Q} \leftarrow$ min-heap event queue
$\Pi_a \leftarrow \emptyset$ for $a \in \{\texttt{left}, \texttt{right}\}$                                 ▷ execution trace
$F(a) \leftarrow 0$ for $a \in \{\texttt{left}, \texttt{right}\}$                          ▷ free time
$L(a) \leftarrow (\mathsf{locked} \leftarrow \text{False},\ \mathsf{chain} \leftarrow \perp)$
**enqueue** all $n_i$ with $|\mathsf{dep}_i| = 0$ as $(\mathsf{start}_i, \mathsf{available}, i)$ into $\mathcal{Q}$

**while** $\mathcal{Q} \neq \emptyset$ **do**
    $(t, \mathsf{type}, i) \leftarrow \mathcal{Q}.\texttt{pop()}$
    **if** $\mathsf{type} = \mathsf{available}$ **then**
        $a^* \leftarrow \texttt{choose\_arm}(F, n_i, L)$
        **Deadlock Check:** If $a^* = \perp \wedge \mathsf{arms}_i = 2 \wedge$ both arms last action is $\mathsf{pick}$
            Determine arm $a'$ matching $\mathsf{source}_i$, delete latest $\mathsf{pick}$ in other arm $a''$
            Remove successors of that $\mathsf{pick}$ from $\mathcal{Q}$ and reheapify
            Schedule $n_i$ at $F(a')$, update both arms, $\Pi_{\texttt{left}}, \Pi_{\texttt{right}}, F, L$
            **enqueue** $(t + \mathsf{take}_i, \mathsf{completed}, i, \mathsf{both})$ and deleted pick
            **continue**
        **if** $a^* = \perp$ **then enqueue** $(t + 1, \mathsf{available}, i)$ **and continue**
        **end if**
        **if** $\mathsf{type}_i = \mathsf{place} \wedge a^* \neq \mathsf{both} \wedge L(a^*).\mathsf{chain} \neq \mathsf{source}_i$ **then**
            **enqueue** $(t + 1, \mathsf{available}, i)$ **and continue**
        **end if**
        **if** $\mathsf{type}_i = \mathsf{pick} \wedge L(a^*).\mathsf{locked}$ **then**
            **enqueue** $(t + 1, \mathsf{available}, i)$ **and continue**
        **else if** $\mathsf{type}_i = \mathsf{pick}$ **then**
            $L(a^*).\mathsf{locked} \leftarrow \text{True},\ L(a^*).\mathsf{chain} \leftarrow \mathsf{target}_i$
        **end if**
        **if** $a^* = \mathsf{both}$ **then**
            $s \leftarrow \max(F(\texttt{left}), F(\texttt{right})),\ e \leftarrow s + \mathsf{take}_i$
            $F(\texttt{left}), F(\texttt{right}) \leftarrow e$
            $\Pi_a \leftarrow \Pi_a \cup \{(s, e, \mathsf{name}_i)\}$ for $a \in \{\texttt{left}, \texttt{right}\}$
            Set $L(\cdot)$ according to which arm held $\mathsf{source}_i$
        **else**
            $s \leftarrow \max(F(a^*), \mathsf{start}_i),\ e \leftarrow s + \mathsf{take}_i$
            $F(a^*) \leftarrow e, \Pi_{a^*} \leftarrow \Pi_{a^*} \cup \{(s, e, \mathsf{name}_i)\}$
        **end if**
        **enqueue** $(e, \mathsf{completed}, i, a^*)$
    **else if** $\mathsf{type} = \mathsf{completed}$ **then**
        Update $L(a)$: release lock if place or $a = \mathsf{both}$ ends chain
        **for all** $j \in \mathsf{succ}_i$ **do**
            $\mathsf{dep}_j \leftarrow \mathsf{dep}_j - 1$
            $\mathsf{start}_j \leftarrow \max(\mathsf{start}_j, t + \mathsf{delay}_j)$
            **if** $\mathsf{dep}_j = 0$ **then enqueue** $(\mathsf{start}_j, \mathsf{available}, j)$
            **end if**
        **end for**
    **end if**
**end while**
**Return:** $\{\Pi_{\texttt{left}}, \Pi_{\texttt{right}}, \max(F(\texttt{left}), F(\texttt{right}))\}$     ▷ output full schedule plan and total time cost

---

```
C8:  stick_on(source="butter", target="bread")(Single arm, 8 seconds)
C9:  place(source="butter", target="table")(Single arm, 8 seconds)
```

Overall Example: DAG nodes output. These node information serve as the input of Stage 2: Graph Re-Traverse for Parallel Optimal Dual-Arm Planning.

```
Nodes:
node_1:
type:  pick
name:  pick(source="table", target="carrots")
arm_num:  1
take_time:  5
edge:  []

node_2:
type:  place
name:  place(source="carrots", target="cutting_board")
arm_num:  1
take_time:  7
edge:  [1]

node_3:
type:  pick
name:  pick(source="table", target="apples")
arm_num:  1
take_time:  5
edge:  []

node_4:
type:  place
name:  place(source="apples", target="cutting_board")
arm_num:  1
take_time:  7
edge:  [3]

node_5:
type:  pick
name:  pick(source="counter", target="knife")
arm_num:  1
take_time:  5
edge:  [2, 4]

node_6:
type:  cut
name:  cut(source="knife", target="carrots")
arm_num:  2
take_time:  10
edge:  [5]

node_7:
type:  cut
name:  cut(source="knife", target="apples")
arm_num:  2
take_time:  10
edge:  [6]

node_8:
type:  place
name:  place(source="knife", target="counter")
arm_num:  1
take_time:  5
edge:  [7]
```

```
node_9:
type:  pick
name:  pick(source="shelf", target="plate")
arm_num:  1
take_time:  5
edge:  []

node_10:
type:  place
name:  place(source="plate", target="table")
arm_num:  1
take_time:  6
edge:  [9]

node_11:
type:  pick
name:  pick(source="table", target="bread")
arm_num:  1
take_time:  5
edge:  [10]

node_12:
type:  place
name:  place(source="bread", target="plate")
arm_num:  1
take_time:  8
edge:  [11]

node_13:
type:  flap_open
name:  flap_open(target="refrigerator")
arm_num:  1
take_time:  3
edge:  []

node_14:
type:  pick
name:  pick(source="refrigerator", target="butter")
arm_num:  1
take_time:  5
edge:  [12, 13]

node_15:
type:  flap_close
name:  flap_close(target="refrigerator")
arm_num:  1
take_time:  3
edge:  [14]

node_16:
type:  stick_on
name:  stick_on(source="butter", target="bread")
arm_num:  1
take_time:  8
edge:  [14]

node_17:
type:  place
name:  place(source="butter", target="table")
arm_num:  1
take_time:  8
edge:  [16]
```

```
node_18:
type:  task_completion
name:  task_completion
arm_num:  0
take_time:  0
edge:  [8, 17]
```

> **Overall Example: RoboPARA output. Final schedule table with dual-arm parallel operation and cross-task step integration.**
>
> ```
> Total execution time:  82 seconds
>
> Left arm schedule table:
> 0-5 pick(source="table", target="carrots")
> 5-12 place(source="carrots", target="cutting_board")
> 12-17 pick(source="counter", target="knife")
> 17-27 cut(source="knife", target="carrots")
> 27-37 cut(source="knife", target="apples")
> 37-42 place(source="knife", target="counter")
> 48-53 pick(source="table", target="bread")
> 53-61 place(source="bread", target="plate")
> 66-69 flap_close(target="refrigerator")
>
> Right arm schedule table:
> 0-5 pick(source="table", target="apples")
> 5-12 place(source="apples", target="cutting_board")
> 17-27 cut(source="knife", target="carrots")
> 27-37 cut(source="knife", target="apples")
> 37-42 pick(source="shelf", target="plate")
> 42-48 place(source="plate", target="table")
> 48-51 flap_open(target="refrigerator")
> 61-66 pick(source="refrigerator", target="butter")
> 66-74 stick_on(source="butter", target="bread")
> 74-82 place(source="butter", target="table")
> ```

## B  X-DAPT Dataset

The X-DAPT dataset is the first benchmark specifically designed to evaluate the parallelism of dual-arm task planning. It contains over 1,000 task instances, each annotated with executable dual-arm action sequences generated by our RoboPARA system. X-DAPT spans a wide range of realistic scenarios including 10 typical scenes, covering tasks closely aligned with everyday human activities.

Each scene is categorized into three levels of difficulty: Easy, Medium, and Hard. Easy packages typically contain fewer than 10 steps and involve basic object manipulations. Medium packages range from 11 to 15 steps and introduce more diverse skills and branching dependencies. Hard packages feature 16 to 20 steps and require complex coordination, including both single-arm and dual-arm manipulations.

For example, in the kitchen scene, easy tasks include "making carrot slices" or "assembling cream bread"; medium tasks involve multistage preparations like "making carrot noodles" or "packing lunchbox"; while hard tasks such as "making stir-fried rice with green peppers" require tightly coordinated dual-arm sequences with logical dependencies and cross-arm tool reuse. Similarly, in the factory setting, tasks span from grasping and placing tools to intricate dual-arm operations like drilling and assembling.

Moreover, the dataset is structured to support DAG-based reasoning, where each task sequence naturally forms a directed acyclic graph with explicit temporal and logical dependencies. This makes X-DAPT a compelling testbed for evaluating not only parallelism and coordination efficiency but also action reordering, causal inference, and plan repair in embodied planning systems.

To the best of our knowledge, X-DAPT is the most comprehensive dataset for dual-arm task planning to date, combining large scale, rich diversity, and fine-grained control signals, enabling rigorous evaluation of real-world robotic planning algorithms. The percentage of packages with different numbers of implementation actions is shown in Fig. 5.

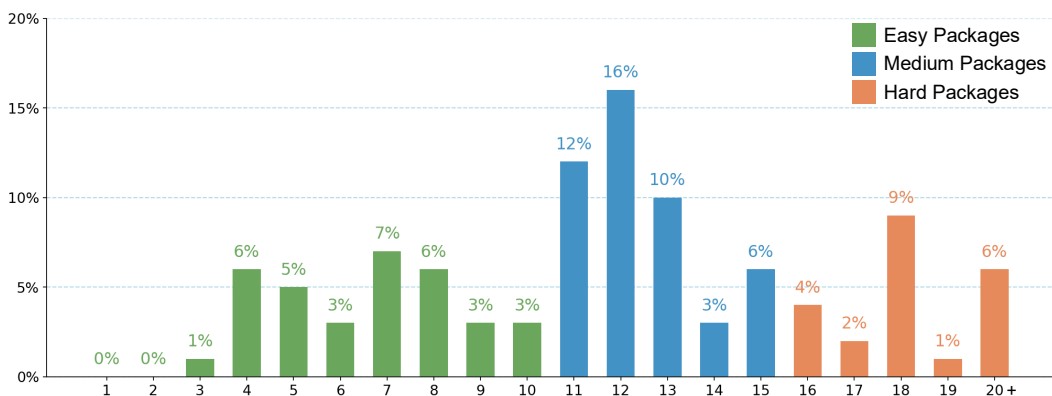

Figure 5: The percentage of packages with different numbers of implementation actions.

## C    SYSTEM-1 / SYSTEM-2 ARCHITECTURAL DECOUPLING

RoboPARA is explicitly designed within the well-established System-1 / System-2 paradigm of embodied AI. In this hierarchy, System-2 is responsible for high-level symbolic planning: it generates global task dependency graphs, reasons about action order and concurrency, and optimizes for dual-arm parallelism. By contrast, System-1 executes these plans at the low-level, handling physical feasibility, motion control, conflict detection, and environment feedback. This division ensures that RoboPARA can focus on logically consistent and parallelizable plans without being encumbered by embodiment details, while System-1 rigorously enforces physical constraints such as grasp validity, collision avoidance, and end-effector differences. Importantly, this decoupling also enables RoboPARA to generalize across heterogeneous dual-arm systems, since the same high-level plan can be executed by different System-1 controllers tailored to specific hardware. In short, RoboPARA contributes a robust System-2 planner that complements existing System-1 execution layers, together forming a complete and feasible dual-arm manipulation system.

## D    COMPARISON OF MAINSTREAM TASK PLANNING METHODS

Mainstream task planning methods can generally be divided into two categories: one is step-by-step or reactive planning, where the robot continuously perceives the environment during execution and generates the next action in real time. This approach is highly flexible and capable of handling unexpected environmental changes, but it lacks a global perspective, is prone to falling into local optima, and tends to be inefficient in tasks with multi-step dependencies or collaboration requirements. The other is global planning, such as the proposed RoboPARA in this paper, which performs unified modeling and optimization of the entire task before execution. This enables systematic consideration of task dependencies and resource constraints, thereby avoiding conflicts and redundant actions, and making it easier to achieve globally optimal or near-optimal execution efficiency. Compared with the former, global planning demonstrates more significant advantages in complex, multi-arm collaborative, and resource-constrained scenarios, greatly enhancing the coherence and predictability of task completion.

# E EXPERIMENTS

## E.1 EXPERIMENTAL SETUP

To evaluate the effectiveness of the RoboPARA framework, we first constructed a comprehensive knowledge base containing all task packages across four distinct scenes: **kitchen**, **office**, **agricultural greenhouse**, and **factory**. Each scene features 15 standardized task packages, categorized by difficulty into **5 easy**, **5 medium**, and **5 hard packages**. To assess performance at different difficulty levels, we design the following test protocol.

For each difficulty level of a selected scene, we create five cumulative task groups:

- Group 1 includes only the 1st task package of this scene;
- Group 2 includes the 1st and 2nd task packages of this scene;
- ...
- Group 5 includes all five task packages of this scene.

For each baseline method (See Sec. E.1.2) and RoboPARA, we conduct experiments across **4 environments** × **3 difficulty levels** × **5 cumulative task groups** = **60 test units**. Final performance metrics—such as TEI, TFR, PPR, and APR (See definitions at Sec. E.1.1) are **averaged over the five task groups within each difficulty level** to ensure robust and representative evaluation.

The characteristics and definition methods of task packages with different difficulty levels can be found in Table 7.

Table 7: Definition, characteristics and detailed implementation of difficulty levels in out experiments.

| Package difficulty | Characteristics | Cumulation of knowledge base |
|---|---|---|
| Easy Packages | Each standard package contains fewer than 10 steps; actions are simple. | Package A
Package A, B
Package A, B, C
Package A, B, C, D
Package A, B, C, D, E |
| Medium Packages | Each standard package contains around 11–15 steps; actions are more complex and diverse. | Package F
Package F, G
Package F, G, H
Package F, G, H, I
Package F, G, H, I, J |
| Hard Packages | Each standard package contains more than 15 steps; actions are more complex and diverse. | Package K
Package K, L
Package K, L, M
Package K, L, M, N
Package K, L, M, N, O |

### E.1.1 EVALUATION METRICS

To provide a comprehensive assessment of the performance, efficiency, and correctness of dual-arm robotic planning, we introduce the following quantitative evaluation criteria:

1) **TEI** (**T**ask **E**fficiency **I**ndex)

$$\text{TEI} = 100 \times \frac{|\mathcal{S}|}{|\mathcal{T}| \cdot T} \tag{9}$$

- $\mathcal{T}$: the set of all scheduled steps (across all packages in the test group)
- $\mathcal{S} \subseteq \mathcal{T}$: the subset of successfully completed steps
- $T$: total task completion time (including planning time, in seconds)

TEI measures the step-level execution efficiency by combining success rate and speed. Higher TEI indicates more steps completed in less time.

2) **TFR** (**T**ask **F**ailure **R**ate)

$$\text{TFR} = 1 - \frac{|\mathcal{S}|}{|\mathcal{T}|} \tag{10}$$

TFR quantifies the proportion of steps that failed during execution, reflecting system robustness. Lower TFR indicates higher reliability and task success.

3) **PPR** (**P**ackage **P**arallelism **R**ate)

$$\text{PPR} = \frac{|\mathcal{S}_{\text{actual}}|}{|\mathcal{S}_{\text{ideal}}|} \tag{11}$$

- $\mathcal{S}_{\text{actual}}$: number of steps actually executed (after step merging and reuse across packages)
- $\mathcal{S}_{\text{ideal}}$: number of steps required if each package is executed independently (i.e., from the knowledge base)

PPR reflects the efficiency gained by merging redundant steps across multiple packages. A higher PPR suggests that steps like "cut with knife" or "place on cutting board" are successfully reused when they appear in multiple packages, instead of executing repeatedly.

4) **APR** (**A**rm **P**arallelism **R**ate)

$$\text{APR} = \sum_{t \in \mathcal{I}} \min\left(\delta_L(t), \delta_R(t)\right) \tag{12}$$

- $\mathcal{I}$: the set of all time intervals where both arms are concurrently active
- $\delta_L(t)$, $\delta_R(t)$: number of actions being executed by the left/right arm at time $t$, respectively
- For dual-arm actions (e.g., `cut` and `stir`), both arms are considered to be active

APR captures the degree of parallel utilization of both robotic arms. Higher APR indicates better concurrency and scheduling efficiency. It rewards not just simultaneous activity but also balanced utilization.

### E.1.2 BASELINES

**LLM**[3] (Wang et al., 2024) introduces a novel conventional task and motion planning framework that utilizes a pre-trained LLM to jointly generate symbolic action sequences and continuous motion parameters, while iteratively refining plans through reasoning over motion failure feedback. The framework operates in a closed-loop fashion: at each planning round, the LLM is prompted with the current state and a trace of past planning feedback to produce a new plan, followed by motion feasibility validation. If failure occurs, categorized feedback (e.g., collision or unreachability) is synthesized and re-fed into the LLM to guide subsequent refinements.

**ChatGPT-Prompts** (Wake et al., 2023) proposes a practical prompting strategy for translating natural language into robot actions using ChatGPT in a few-shot setup. The system receives human instructions and environment descriptions, then generates structured JSON outputs detailing robot actions and environment updates. It supports multi-step task planning by reusing post-action environment states, thereby mitigating the token limit issue.

**VOYAGER** (Wang et al., 2023) is the first lifelong embodied agent powered by GPT-4 in Minecraft. It autonomously explores, learns skills, and builds a compositional skill library without human intervention. VOYAGER operates in a ReAct-style loop with three innovations: (1) an automatic curriculum that proposes increasingly complex tasks to drive exploration, (2) a skill library storing reusable action programs, and (3) an iterative prompting mechanism that incorporates environment feedback, execution errors, and GPT-4 self-verification for robust code refinement. VOYAGER outperforms ReAct (Yao et al., 2023), Reflexion (Shinn et al., 2023), and AutoGPT (aut, 2023) across all metrics.

**Embodied TaPA** (Wu et al., 2023) introduces an LLM-based embodied task planner that grounds action plans in realistic physical scenes using a multi-view object perception pipeline. It collects RGB images from multiple viewpoints, applies open-vocabulary detection to generate an object list, and conditions GPT-3.5 to produce executable plans.

**LLM-Planner** (Song et al., 2023) performs few-shot high-level planning for embodied agents using GPT-3, grounded in the agent's current observation. Given an instruction and detected objects from the scene, it generates a sequence of subgoals using in-context prompting with retrieved examples. The planner executes one round of goal prediction, followed by execution via a separate low-level controller. If a subgoal fails due to timeout or invalid actions, the agent re-queries the environment and replans grounded on updated visible objects.

**FLTRNN** (Zhang et al., 2024) addresses the faithfulness problem in long-horizon task planning with LLMs by introducing a memory-augmented, language-based RNN framework. Given a high-level task goal, it first decomposes the instruction into subgoals using a pretrained LLM, then solves each subtask sequentially with long-short term memory to retain relevant rules and execution summaries. The model incorporates a Rule Chain-of-Thought mechanism and an external memory graph to enhance reasoning and prevent rule violations.

**RoCo** (Mandi et al., 2023) introduces a multi-robot collaboration framework driven by LLM-based dialogic planning and motion execution. Each robot is equipped with an LLM agent that decomposes high-level tasks into spatially grounded subtasks through natural language communication. Robots engage in multi-turn dialogue to allocate goals, reason over constraints, and jointly refine execution plans. After each subtask is assigned, motion plans are generated and validated; execution failures trigger replanning via updated dialogue rounds.

For all the baselines, we implemented their respective methods and adapted the specific tasks in their prompts to fit our scenes. For detailed prompts of the baselines, please refer to our codebase in supplementary materials.

### E.2 BACKBONE MODELS

We adopt m3e-base (Team) (`https://github.com/moka-ai/m3e`) as the retriever of RAG.

To evaluate RoboPARA, we conducted the full set of experiments on top of two LLM backbones—OpenAI ChatGPT-4o (OpenAI) (`https://chatgpt.com/`) and DeepSeek V3 (dee) (`https://www.deepseek.com/`).

### E.3 ABLATIONS

We ablate four design choices (environment information, instructional constraints, arm selector, and DAG correction) in VOYAGER and study their impact on exploration performance.

- **w/o environment information:** We exclude environment information from the prompt for DAG generation.

- **w/o instructional constraints:** We exclude instructional constraints 2) to 18) from the prompt for the first call of DAG generation. See full system prompt for DAG stage (First Call) in Sec. A.2.2 for details.

- **w/o arm selector deadlock check and rebuild:** We only apply Case IV (basic selecting strategy: choose the earliest free arm) in Alg. 3, excluding arm state check, deadlock check and rescheduling.

- **w/o DAG correction:** We exclude checking structural consistency and rebuilding DAG graph in Alg. 1, only remaining the first call.

### E.4 MORE RESULTS ON DIFFERENT LLM FOUNDATIONS

The main text presents experimental results based on the GPT-4o foundation model. Here, we additionally report results of the full set of experiments under the DeepSeek V3 foundation model. Across both LLM backbones, our framework consistently outperforms all baselines, highlighting its robustness to variations in underlying models.

Table 8: **Quantitative results of dual-arm task planning under 4 scenes with DeepSeek V3 foundation model.** ↑: higher is better, ↓: lower is better. The red, orange, and yellow colors denote the best, the second best, and the third best results, respectively.

| Scenes | Kitchen | | | | Office | | | | Agricultural Greenhouse | | | | Factory | | | |
|---|---|---|---|---|---|---|---|---|---|---|---|---|---|---|---|---|
| | TEI ↑ | TFR ↓ | PPR ↑ | APR ↑ | TEI ↑ | TFR ↓ | PPR ↑ | APR ↑ | TEI ↑ | TFR ↓ | PPR ↑ | APR ↑ | TEI ↑ | TFR ↓ | PPR ↑ | APR ↑ |
| LLM[3] | 0.934 | 0.467 | 0.000 | 0.129 | 1.572 | 0.142 | 0.000 | 0.222 | 1.171 | 0.179 | 0.000 | 0.190 | 1.260 | 0.351 | 0.000 | 0.235 |
| ChatGPT-Prompts | 0.762 | 0.200 | 0.000 | 0.000 | N/A | 0.208 | 0.000 | 0.000 | N/A | 0.194 | 0.000 | 0.000 | N/A | 0.450 | 0.000 | 0.000 |
| VOYAGER | 0.000 | 1.000 | 0.000 | 0.000 | 0.000 | 1.000 | 0.000 | 0.000 | 0.000 | 1.000 | 0.000 | 0.000 | 0.000 | 1.000 | 0.000 | 0.000 |
| Embodied TaPA | 0.858 | 0.479 | 0.000 | 0.059 | 1.137 | 0.461 | 0.000 | 0.094 | 0.906 | 0.553 | 0.000 | 0.091 | 0.687 | 0.467 | 0.000 | 0.027 |
| LLM-Planner | 0.956 | 0.304 | 0.004 | 0.060 | 1.038 | 0.490 | 0.000 | 0.039 | 0.951 | 0.431 | 0.000 | 0.094 | 0.885 | 0.356 | 0.000 | 0.043 |
| FLTRNN | 0.912 | 0.000 | 0.000 | 0.000 | 1.072 | 0.050 | 0.000 | 0.000 | 0.921 | 0.057 | 0.000 | 0.000 | 0.875 | 0.162 | 0.000 | 0.004 |
| RoCo | 1.001 | 0.084 | 0.000 | 0.045 | 1.145 | 0.104 | 0.000 | 0.057 | 0.933 | 0.159 | 0.000 | 0.101 | 0.905 | 0.043 | 0.000 | 0.063 |
| **RoboPARA** | 1.302 | 0.063 | 0.124 | 0.297 | 1.659 | 0.000 | 0.030 | 0.302 | 1.251 | 0.005 | 0.000 | 0.319 | 1.445 | 0.008 | 0.000 | 0.373 |

Table 9: **Quantitative results of dual-arm task planning with different package difficulties with DeepSeek V3 foundation model.** ↑: higher is better, ↓: lower is better. The red, orange, and yellow colors denote the best, the second best, and the third best results, respectively.

| Package difficulty | Easy Packages | | | | Medium Packages | | | | Hard Packages | | | |
|---|---|---|---|---|---|---|---|---|---|---|---|---|
| | TEI ↑ | TFR ↓ | PPR ↑ | APR ↑ | TEI ↑ | TFR ↓ | PPR ↑ | APR ↑ | TEI ↑ | TFR ↓ | PPR ↑ | APR ↑ |
| LLM[3] | 1.869 | 0.125 | 0.000 | 0.238 | 1.225 | 0.327 | 0.000 | 0.192 | 0.609 | 0.401 | 0.000 | 0.152 |
| ChatGPT-Prompts | N/A | 0.227 | 0.000 | 0.000 | N/A | 0.131 | 0.000 | 0.000 | N/A | 0.430 | 0.000 | 0.000 |
| VOYAGER | 0.000 | 1.000 | 0.000 | 0.000 | 0.000 | 1.000 | 0.000 | 0.000 | 0.000 | 1.000 | 0.000 | 0.000 |
| Embodied TaPA | 1.482 | 0.265 | 0.000 | 0.099 | 0.774 | 0.591 | 0.000 | 0.052 | 0.435 | 0.614 | 0.000 | 0.053 |
| LLM-Planner | 1.630 | 0.210 | 0.003 | 0.080 | 0.889 | 0.298 | 0.000 | 0.077 | 0.354 | 0.678 | 0.000 | 0.020 |
| FLTRNN | 1.406 | 0.000 | 0.000 | 0.000 | 0.930 | 0.087 | 0.000 | 0.000 | 0.499 | 0.115 | 0.000 | 0.003 |
| RoCo | 1.551 | 0.083 | 0.000 | 0.117 | 0.958 | 0.061 | 0.000 | 0.043 | 0.479 | 0.148 | 0.000 | 0.040 |
| **RoboPARA** | 2.053 | 0.000 | 0.047 | 0.309 | 1.385 | 0.000 | 0.013 | 0.333 | 0.787 | 0.057 | 0.056 | 0.327 |

## E.5 OTHER EXPERIMENTS

### E.5.1 ADDITIONAL RESULTS ON OTHER SCENES

Due to space constraints, the main text reports results from only four scenarios. Here, we provide the complete test results for the remaining six scenarios in the X-DAPT dataset.

### E.5.2 ADDITIONAL RESULTS ON ROBOTWIN DATASET

To further test the generalization and robustness of our planning method across existing datasets, we extend our experiments to the RoboTwin (Mu et al., 2025) dataset, which provides 50 dual-arm manipulation tasks of varying difficulty. While RoboTwin is originally designed for validating low-level execution (System-1), its tasks are inherently atomic and short-horizon. To align with our focus on high-level dual-arm planning, we reorganize RoboTwin by grouping and integrating multiple related tasks into coherent scene-level configurations. This result in the following dataset in Table 12.

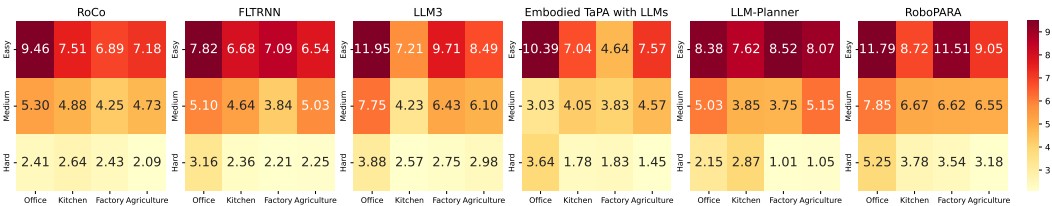

Figure 6: **TEI performance across different frameworks with DeepSeek V3 foundation model.** The heatmap shows the total TEI scores (higher is better) aggregated over all packages for each combination of scene (horizontal axis) and package difficulty level (vertical axis). RoboPARA consistently achieves the highest TEI across all scenarios, particularly excelling in harder packages and complex scenes. VOYAGER (Wang et al., 2023) and ChatGPT-Prompts (Wake et al., 2023) are not included due to incompatibility.

Table 10: Quantitative results on the **Supermarket**, **Hospital**, and **Disaster Rescue** scene. ↑: higher is better, ↓: lower is better. The red, orange, and yellow colors denote the best, the second best, and the third best results, respectively.

| Scene | Supermarket Scene | | | | Hospital Scene | | | | Disaster Rescue Scene | | | |
|---|---|---|---|---|---|---|---|---|---|---|---|---|
| | TEI ↑ | TFR ↓ | PPR ↑ | APR ↑ | TEI ↑ | TFR ↓ | PPR ↑ | APR ↑ | TEI ↑ | TFR ↓ | PPR ↑ | APR ↑ |
| LLM3 | 0.81 | 0.33 | 0.00 | 0.00 | 0.94 | 0.25 | 0.00 | 0.01 | 0.87 | 0.24 | 0.00 | 0.00 |
| ChatGPT-Prompts | 1.02 | 0.02 | 0.00 | 0.01 | 0.83 | 0.11 | 0.00 | 0.00 | 0.79 | 0.12 | 0.00 | 0.01 |
| VOYAGER | 0.00 | 1.00 | 0.00 | 0.00 | 0.00 | 1.00 | 0.00 | 0.00 | 0.00 | 1.00 | 0.00 | 0.00 |
| Embodied TaPA | 0.85 | 0.20 | 0.00 | 0.01 | 0.76 | 0.08 | 0.00 | 0.01 | 0.91 | 0.18 | 0.00 | 0.01 |
| LLM-Planner | 1.21 | 0.20 | 0.00 | 0.16 | 1.08 | 0.21 | 0.00 | 0.14 | 1.10 | 0.22 | 0.00 | 0.15 |
| FLTRNN | 1.05 | 0.00 | 0.00 | 0.01 | 1.03 | 0.02 | 0.00 | 0.02 | 1.04 | 0.00 | 0.00 | 0.01 |
| RoCo | 1.01 | 0.15 | 0.00 | 0.01 | 0.95 | 0.19 | 0.00 | 0.01 | 1.01 | 0.16 | 0.00 | 0.02 |
| RoboPARA | 1.34 | 0.00 | 0.12 | 0.39 | 1.29 | 0.00 | 0.00 | 0.41 | 1.35 | 0.01 | 0.17 | 0.43 |

Table 11: Quantitative results on the **Hotel**, **Pet Shop**, and **Library** scene. ↑: higher is better, ↓: lower is better. The red, orange, and yellow colors denote the best, the second best, and the third best results, respectively.

| Scene | Hotel Scene | | | | Pet Shop Scene | | | | Library Scene | | | |
|---|---|---|---|---|---|---|---|---|---|---|---|---|
| | TEI ↑ | TFR ↓ | PPR ↑ | APR ↑ | TEI ↑ | TFR ↓ | PPR ↑ | APR ↑ | TEI ↑ | TFR ↓ | PPR ↑ | APR ↑ |
| LLM3 | 0.88 | 0.26 | 0.00 | 0.01 | 0.91 | 0.22 | 0.00 | 0.01 | 0.89 | 0.28 | 0.00 | 0.01 |
| ChatGPT-Prompts | 0.93 | 0.19 | 0.00 | 0.01 | 0.86 | 0.00 | 0.00 | 0.00 | 0.92 | 0.12 | 0.00 | 0.01 |
| VOYAGER | 0.00 | 1.00 | 0.00 | 0.00 | 0.00 | 1.00 | 0.00 | 0.00 | 0.00 | 1.00 | 0.00 | 0.00 |
| Embodied TaPA | 0.84 | 0.00 | 0.00 | 0.01 | 0.82 | 0.25 | 0.00 | 0.01 | 0.87 | 0.23 | 0.00 | 0.01 |
| LLM-Planner | 1.09 | 0.23 | 0.00 | 0.16 | 1.07 | 0.19 | 0.00 | 0.15 | 1.13 | 0.18 | 0.00 | 0.15 |
| FLTRNN | 1.00 | 0.00 | 0.00 | 0.02 | 1.02 | 0.00 | 0.00 | 0.02 | 1.06 | 0.01 | 0.00 | 0.02 |
| RoCo | 0.97 | 0.18 | 0.00 | 0.01 | 0.94 | 0.16 | 0.00 | 0.01 | 0.98 | 0.21 | 0.00 | 0.01 |
| RoboPARA | 1.12 | 0.00 | 0.00 | 0.22 | 1.36 | 0.00 | 0.05 | 0.24 | 1.32 | 0.00 | 0.04 | 0.45 |

We compare RoboPARA with the baselines on the extended RoboTwin dataset, as shown in Table 13. Across all difficulty levels, RoboPARA consistently generates valid and executable plans while most baselines suffer from frequent task failures in more complex scenarios. Moreover, RoboPARA demonstrates stronger utilization of dual-arm concurrency, leading to higher efficiency and smoother coordination between arms. These results underscore RoboPARA's generalizability and robustness across diverse planning domains and datasets.

## E.6 CLOSED-LOOP ERROR HANDLING

Closed-loop capability is recognized as a cornerstone of embodied AI systems. Unlike open-loop pipelines, where planning is executed once without feedback, closed-loop systems can perceive the environment, detect failures, and dynamically replan. This ensures robustness when execution diverges from the original plan due to uncertainties in perception, manipulation, or external disturbances. RoboPARA operates at the System-2 level of high-level planning, yet it is explicitly designed with mechanisms to respond to System-1 execution contingencies. In practice, when unexpected physical realities occur (such as an object being dropped, misplaced, or obstructed), RoboPARA can immediately update its task description with the new environment state and generate a revised plan.

To make this process concrete, we provide the following illustrative example: the robot is tasked with preparing two simple dishes, "make carrot slices (A1–A5)" and "make apple salad (B1–B5)". The step-by-step breakdown is presented in Table 14 and Table 15.

RoboPARA then generates the execution plan, as shown in Table 16.

We assume that at the 12-second mark, the robot's right arm accidentally drops the apple onto the table while attempting to place it on the cutting board, interrupting the task. RoboPARA is capable of updating the prompt, re-executes the planning process, and generates a new plan starting from that moment. It can be observed that RoboPARA successfully resumes and completes the task. The new plan from that moment is presented in the following Table 17. Despite these disturbances, RoboPARA demonstrates the ability to replan by updating the latest observations. This design enables seamless

Figure 7: **TEI performance across different frameworks with GPT-4o foundation model.** The heatmap shows the total TEI scores (higher is better) aggregated over all packages for each combination of scene (horizontal axis) and package difficulty level (vertical axis). RoboPARA consistently achieves the highest TEI across all scenarios, particularly excelling in harder packages and complex scenes. VOYAGER (Wang et al., 2023) is not included due to incompatibility.

Table 12: Complete list of the reorganized RoboTwin dataset for high-level planning.

| Difficulty | Task ID(s) |
|---|---|
| Easy | 3, 4, 7 |
| Medium | 24, 26, 26–27, 28–30, 29–31 |
| Hard | 1–6, 8–10, 22–24, 32–34, 36–37, 38–39, 48–50 |

continuity between planning and replanning, reinforcing RoboPARA's robustness across dynamic environments.

Building on this illustrative case, we further conduct a broader set of evaluations to systematically assess the robustness of RoboPARA's closed-loop replanning capability. Specifically, we select nine representative tasks of varying difficulty levels from the X-DAPT kitchen dataset. For each task, we introduce artificially designed execution errors to simulate realistic System-1 failures. Despite these disturbances, RoboPARA consistently replan and complete the tasks, confirming its ability to adapt dynamically and sustain reliable performance under error-prone conditions. The summarized settings and results are presented in Table 18.

### E.7    FIELD TESTS

We select several representative samples from the benchmark dataset in our experiments for real-world testing. The hardware setups and representative testing environments of the three dual-arm robotic platforms are shown in Fig. 8.

### E.7.1    HUMANOID ROBOT

The results of the real-world tests on Humanoid robot are presented in Fig. 1. The results are based on GPT-4o foundation LLM.

Table 13: Quantitative results on the RoboTwin dataset. ↑: higher is better, ↓: lower is better. The  red ,  orange , and  yellow  colors denote the best, the second best, and the third best results, respectively.

| Method | Easy | | | Medium | | | Hard | | |
|---|---|---|---|---|---|---|---|---|---|
| | TEI ↑ | TFR ↓ | APR ↑ | TEI ↑ | TFR ↓ | APR ↑ | TEI ↑ | TFR ↓ | APR ↑ |
| LLM3 | 6.7 | 0.0 | 0.0 | 2.1 | 0.2 | 0.1 | 1.0 | 0.4 | 0.1 |
| ChatGPT-Prompts | 8.0 | 0.0 | 0.3 | 3.1 | 0.0 | 0.2 | 1.7 | 0.0 | 0.1 |
| Embodied TaPA | 8.0 | 0.0 | 0.3 | 4.5 | 0.0 | 0.2 | 1.8 | 0.0 | 0.2 |
| LLM-Planner | 6.7 | 0.0 | 0.1 | 2.2 | 0.2 | 0.1 | 1.0 | 0.4 | 0.1 |
| FLTRNN | 6.7 | 0.0 | 0.0 | 3.1 | 0.0 | 0.2 | 1.8 | 0.0 | 0.1 |
| RoCo | 7.3 | 0.0 | 0.2 | 3.1 | 0.0 | 0.2 | 1.4 | 0.2 | 0.1 |
| RoboPARA | 10.6 | 0.0 | 0.5 | 5.2 | 0.0 | 0.6 | 2.4 | 0.0 | 0.3 |

Table 14: Detailed breakdown of the task "make carrot slices"

| Step | Action | Source | Target | Arm Type | Duration |
|------|--------|--------|--------|----------|----------|
| A1 | pick | table | carrots | Single arm | 5 sec |
| A2 | place | carrots | cutting_board | Single arm | 7 sec |
| A3 | pick | counter | knife | Single arm | 5 sec |
| A4 | cut | knife | carrots | Dual arm | 10 sec |
| A5 | place | knife | counter | Single arm | 5 sec |

Table 15: Detailed breakdown of the task "make apple salad"

| Step | Action | Source | Target | Arm Type | Duration |
|------|--------|--------|--------|----------|----------|
| B1 | pick | table | apples | Single arm | 5 sec |
| B2 | place | apples | cutting_board | Single arm | 7 sec |
| B3 | pick | counter | knife | Single arm | 5 sec |
| B4 | cut | knife | apples | Dual arm | 10 sec |
| B5 | place | knife | counter | Single arm | 5 sec |

### E.7.2 FRANKA RESEARCH 3 STATION

We perform real-world experiments using the Franka robotic arms in agricultural greenhouse scene with DeepSeek V3 foundation LLM. The task is defined as "**pack cucumbers and prepare seed tray for sprouting**", and all baselines are evaluated under this task.

---

Retrieved packages and steps for task "pack cucumbers and prepare seed tray for sprouting".

```
Package A: Harvest and Pack Cucumbers
A1:  pick(source="storage_shelf", target="cucumber")(Single arm, 5
seconds)
A2:  place(source="cucumber", target="basket")(Single arm, 6 seconds)
A3:  pick(source="counter", target="knife")(Single arm, 5 seconds)
A4:  cut(source="knife", target="cucumber_stem")(Dual arm, 10 seconds)
A5:  place(source="knife", target="counter")(Single arm, 5 seconds)

Package B: Prepare Seed Tray for Sprouting
B1:  pick(source="storage_shelf", target="seed_tray")(Single arm, 5
seconds)
B2:  place(source="seed_tray", target="work_table")(Single arm, 6
seconds)
B3:  pick(source="seed_bin", target="seeds")(Single arm, 5 seconds)
B4:  pour_into(source="seeds", target="seed_tray")(Single arm, 7
seconds)
B5:  place(source="lettuce_seeds", target="seed_bin")(Single arm, 5
seconds)
```

---

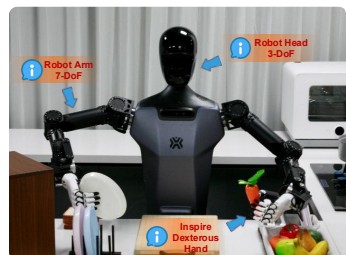 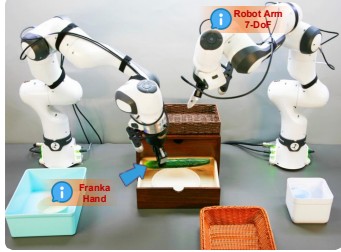 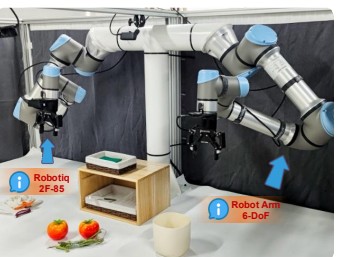

   (a) Humanoid Robot       (b) Franka Research 3 Station       (c) UR5e Station

Figure 8: Hardware configurations of the dual-arm robots used in the experiments.

Table 16: Original execution plan generate by RoboPARA.

| Time (s) | Left Arm Action | Right Arm Action |
| --- | --- | --- |
| 0-5 | pick("table", "carrots") | pick("table", "apples") |
| 5-12 | place("carrots", "cutting_board") | place("apples", "cutting_board") |
| 12-17 | pick("counter", "knife") | - |
| 17-27 | cut("knife", "carrots") | cut("knife", "carrots") |
| 27-37 | cut("knife", "apples") | cut("knife", "apples") |
| 37-42 | place("knife", "counter") | - |

Table 17: The new plan generated by RoboPARA at the 12-second mark.

| Time (s) | Left Arm Action | Right Arm Action |
| --- | --- | --- |
| 0-5 | - | pick("table", "apples") |
| 5-7 | - | place("apples", "cutting_board") |
| 7-12 | pick("counter", "knife") | place("apples", "cutting_board") |
| 12-22 | cut("knife", "carrots") | cut("knife", "carrots") |
| 22-32 | cut("knife","apples") | cut("knife", "apples") |
| 32-37 | place("knife", "counter") | - |

```
B6:  pick(source="water_station", target="watering_can")(Single arm, 5
seconds)
B7:  pour_into(source="watering_can", target="seed_tray")(Single arm,
8 seconds)
B8:  place(source="watering_can", target="water_station")(Single arm,
5 seconds)
```

The initial setup of the scene is shown in Fig. 9. Planning results of RoboPARA and all baselines are shown in Fig. 10. RoboPARA enables reordering and parallel execution while maintaining correctness by uncovering latent logical dependencies between task steps, thereby achieving optimal parallelism and outperforming all baseline methods (Table 19).

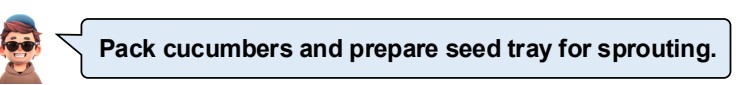
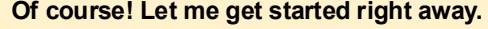
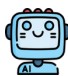
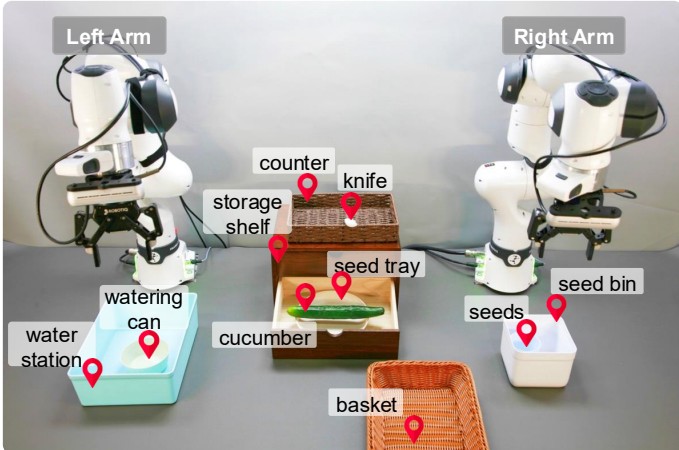

Figure 9: **Initial setup of the agricultural greenhouse scene for RoboPARA and all baselines.** Two Franka Research 3 robotic arms are used in this experiment.

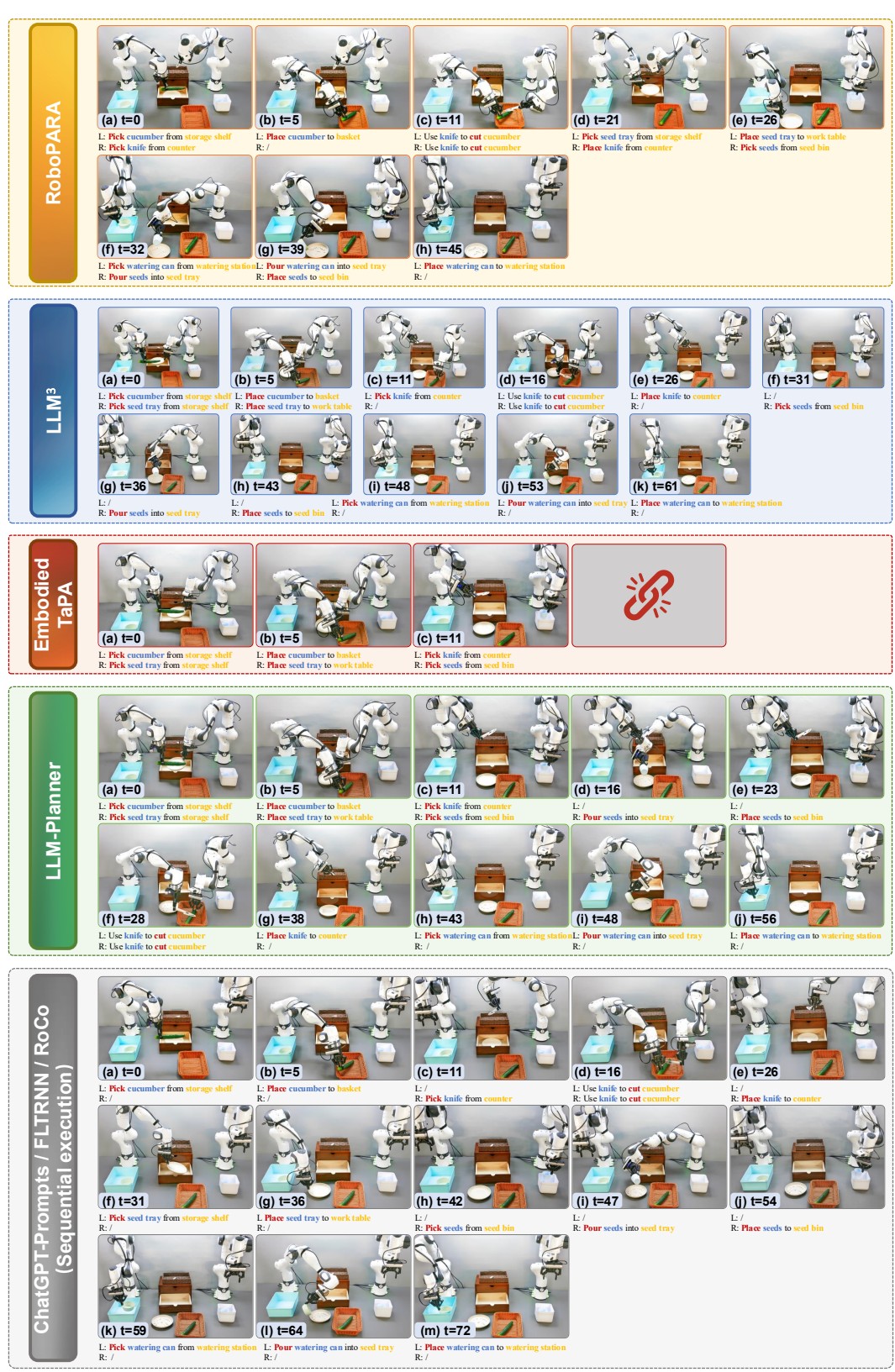

Figure 10: **Real-world experiments using the Franka robotic arms in agricultural greenhouse scene.** L and R denote the left and right arms, respectively. VOYAGER (Wang et al., 2023) is not included because it cannot provide explicit plans. Embodied TaPA (Wu et al., 2023) fails because it generates a next action (cutting the cucumber) that requires both arms, while the right arm is still occupied. RoboPARA outperforms all baseline methods.

Table 18: Closed-loop error handling evaluation. RoboPARA successfully replan in all cases, demonstrating robust recovery across various difficulty levels.

| Difficulty | Task | Execution Error | Replan Success |
|---|---|---|---|
| Easy | make carrot and apple slices | apple fell off the cutting board | ✓ |
| Easy | make cream bread | bread fell off the plate | ✓ |
| Easy | set the table | fork fell while taking it out of the cabinet | ✓ |
| Medium | make butter spinach and tomato sandwich | butter fell when placing it back to the counter | ✓ |
| Medium | make fruit salad | banana fell while getting it from the fridge | ✓ |
| Medium | assemble a sandwich | bread fell off the plate | ✓ |
| Hard | make seaweed mushroom soup | mushroom fell while picking it up from the cutting board | ✓ |
| Hard | make creamy pasta | cheese fell while taking it out of the fridge | ✓ |
| Hard | make stir-fried rice with green peppers | knife slipped while cutting the green pepper | ✓ |

Table 19: **Comparison between RoboPARA and baselines using Franka robotic arms in a real-world agricultural greenhouse task with DeepSeek V3 foundation LLM.** RoboPARA exhibits the best parallelism and task completion rate, highlighting its high efficiency and strong adaptability to specific application scenarios.

| Package difficulty | Total execution time (seconds) | Execution success rate | Number of parallel intervals |
|---|---|---|---|
| LLM[3] (Wang et al., 2024) | 66 | 100.0% | 3 |
| ChatGPT-Prompts (Wake et al., 2023) | 77 | 100.0% | 1 |
| VOYAGER (Wang et al., 2023) | - | - | - |
| Embodied TaPA (Wu et al., 2023) | 11 (Fail) | 46.2% (6 of total 13 steps) | 3 (Fail) |
| LLM-Planner (Song et al., 2023) | 61 | 100.0% | 4 |
| FLTRNN (Zhang et al., 2024) | 77 | 100.0% | 1 |
| RoCo (Mandi et al., 2023) | 77 | 100.0% | 1 |
| **RoboPARA** | 50 | 100.0% | 6 |

### E.7.3 UR5E STATION

We conduct real-world validation across 4 different test scenes (kitchen, office, agricultural greenhouse, factory) using two UR5e robotic arms. The initial setup of the four scenes are illustrated in Fig. 11. The corresponding results are presented in Fig. 12, Fig. 13, Fig. 14, and Fig. 15 respectively. The results are based on GPT-4o foundation LLM.

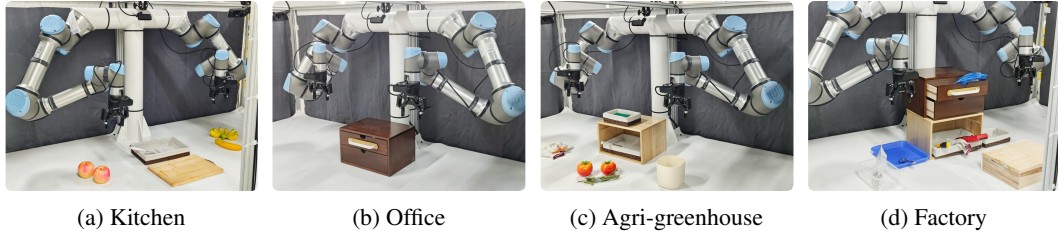

(a) Kitchen     (b) Office     (c) Agri-greenhouse     (d) Factory

Figure 11: Initial setup of the 4 scenes for real-world experiments upon UR5e station.

## LLM USAGE STATEMENT

Large Language Models (LLMs) are used in this work only for grammar checking and light language refinement of certain parts of the INTRODUCTION and CONCLUSION sections to improve readability and presentation quality. All core contributions, including the design of the method, theoretical formulations, experimental setup, and analysis are independently conceived and implemented by the authors without the involvement of LLMs. The full implementation code, experimental scripts and datasets are prepared entirely by the authors, ensuring the accuracy and reproducibility of the reported results. The authors take full responsibility for the limited use of LLMs in language polishing and for all claims made in this paper.

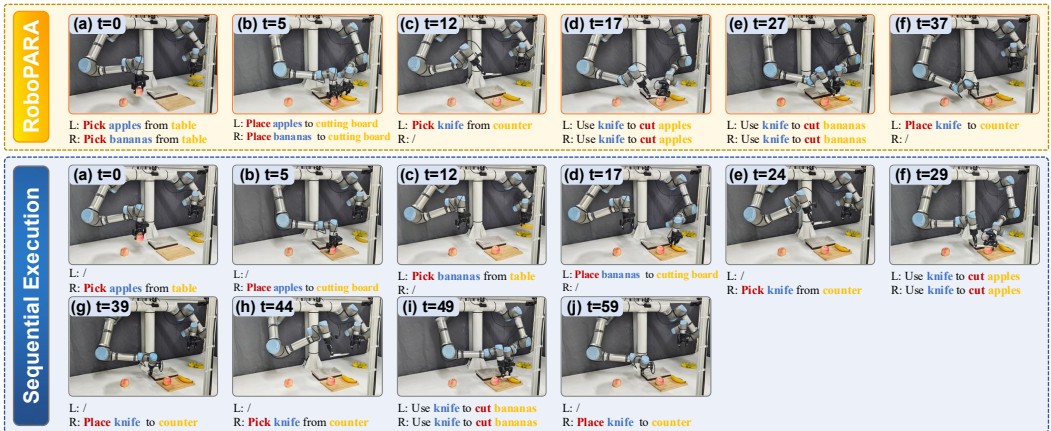

Figure 12: **Real-world experiments in kitchen scene using the UR5e robotic arms.** The task is "cut apples and bananas". Compared with sequential execution, RoboPARA excels at **step consolidation and action merging**, leading to reduced execution time. This improvement is enabled by the DAG-based dependency reasoning (Sec. 3.4), which accurately captures inter-step dependencies and reveals optimal execution paths.

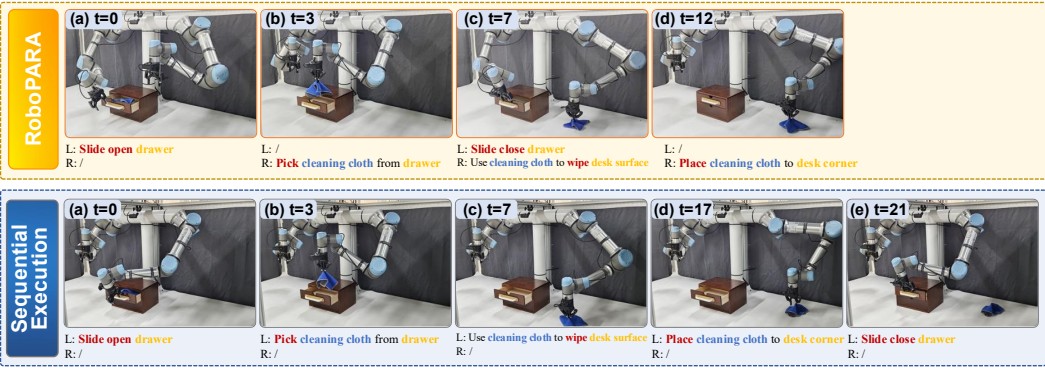

Figure 13: **Real-world experiments in office scene using the UR5e robotic arms.** The task is "clean the office desk". Compared with sequential execution, RoboPARA emphasizes **real-world dual-arm collaboration that closely resembles human behavior**. For example, a person typically opens a drawer with one hand and picks an item with the other, rather than using the same hand to open, fetch, and close the drawer sequentially.

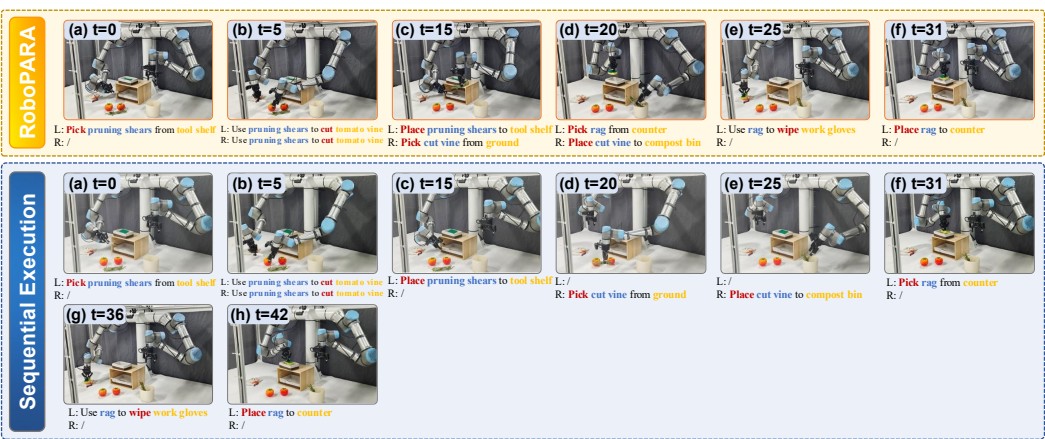

Figure 14: **Real-world experiments in agricultural greenhouse scene using the UR5e robotic arms.** The task is "cut and sort tomato vines". Compared with sequential execution, RoboPARA emphasizes **intra-package parallelism**, which fully leverages dual-arm resources and greatly boosts task completion efficiency.

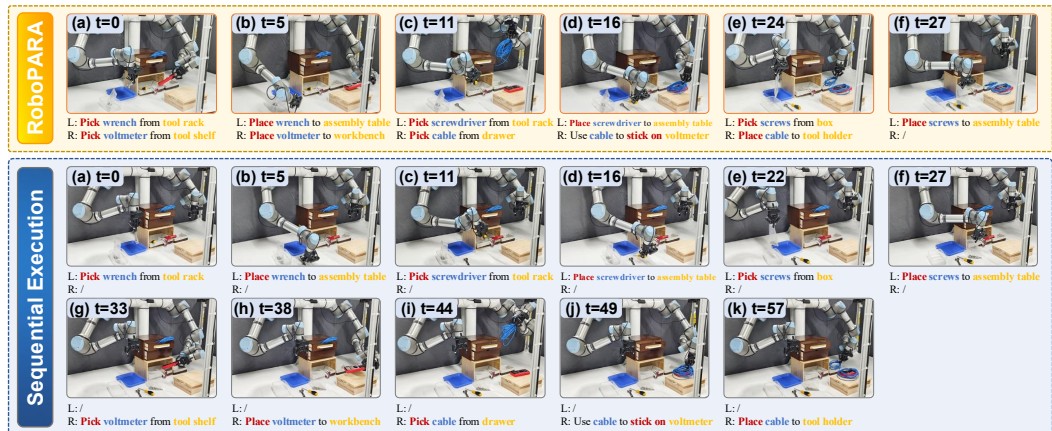

Figure 15: **Real-world experiments in factory scene using the UR5e robotic arms.** The task is "assemble tool components and prepare electrical tools". Compared with sequential execution, RoboPARA highlights **cross-package parallelism**, demonstrating higher-level task planning and integration capabilities.

