# OpenReview forum: "RoboPARA: Dual-Arm Robot Planning with Parallel Allocation and Recomposition Across Tasks"
_ICLR.cc/2026/Conference — ICLR 2026 Poster_

### Official Review · Reviewer_4Bah · 2025-10-29

**Soundness:** 2
**Presentation:** 2
**Contribution:** 2
**Rating:** 6
**Confidence:** 4

**Summary:**

This work proposes RoboPARA, a framework for using large language models (LLMs) for dual-arm task parallelism planning. The framework employs a two-stage process: (1) Dependency Graph-based Planning Candidates Generation and (2) Graph Re-Traversal-based Dual-Arm Parallel Planning. Highlight of the proposed framework is constructing and traversing directed acyclic graphs (DAGs) that model task dependencies and eliminate redundancy. The authors also introduce the Cross-Scenario Dual-Arm Parallel Task dataset (X-DAPT dataset), designed to evaluate dual-arm task parallelism across diverse scenarios and difficulty levels. Extensive empirical results are provided to demonstrate that RoboPARA significantly outperforms existing planning methods, especially in complex task combinations and in requiring less execution time than baseline methods while achieving similar task performance.

**Strengths:**

1. The paper has high presentation quality and easy-to-follow writing. The appendix section provides extensive details on various aspects of the method details and experiment settings.

2. Real world experiments span multiple robot setup and various tasks, which speak for the generality of the proposed framework and applicability to real robot systems.

3. The proposed framework achieves clear reduction in execution time while maintaining task success. Comprehensive experimental results and dedicated efforts to re-implement and compare against various baselines, which had different assumptions and task settings in the original works.

**Weaknesses:**

1. Lack of video results. Although various figures throughout the manuscript show task setups and robot configurations, a video of actual task completion will be much more convincing and conveys more clarity on the task settings.

2. Minor grammar issues, e.g. lack of spacing (Line 1834, 1835, RoboPARAsuccessfully, RoboPARAdemonstrates).

3. The proposed framework treats bimanual manipulation as a planning problem over two arms, but concurrently there is also many efforts on learning end-to-end policies from bimanual teleoperation data. It is unclear whether the two-arm planning approach is still needed or would need a major modification when bimanual policies can easily leverage two arms at once to perform many complex task motions.

**Questions:**

1. The authors mentioned "GPT-4o and DeepSeek V3 APIs incurs significant costs" -- besides financial costs, does using the LLM APIs incur delays in between queries, and does that pose a problem for smooth robot action execution?

2. What kind of inaccuracies do the robot agents struggle to self-correct or re-plan? Again, providing video results would be very helpful.

---

> ### Author Response · Authors · 2025-11-16
> **Author Response to Reviewer 4Bah (Part 1)**
>
> **W1: Lack of video results.**
>
> We appreciate this comment. Please check the video demos in the latest uploaded supplementary materials.
>
> **W2: Minor grammar and formatting.**
>
> Thanks for pointing it out! This has been fixed in the most recently uploaded version.
>
> **W3: Unclear necessity of explicit two-arm planning given emerging end-to-end bimanual policy learning.**
>
> We thank the reviewer for raising this important point. Our adoption of a structured dual-arm planning framework is a deliberate design choice, motivated by the goals of **reliability, stability, controllability, and scalability**, all of which are essential for long-horizon execution in real-world robotic systems.
>
> 1. **Determinism and constraint control:** Current end-to-end bimanual policies couple perception and reasoning, producing non-deterministic outputs that are difficult to verify. In contrast, RoboPARA explicitly separates symbolic reasoning (System-2) from low-level perception and control (System-1). This separation provides full transparency of the dependency graph, enables strict enforcement of task ordering, and allows reliable constraint checking, all of which are essential for the safety and synchronization of dual-arm execution.
> 2. **Safety** **and** **consistency****:** Dual-arm manipulation requires precise lock management and collision-free scheduling. Existing end-to-end bimanual policies lack explicit modeling of arm exclusivity, object locking, or deadlock prevention, and their implicit visual reasoning cannot guarantee satisfaction of safety-critical constraints. RoboPARA’s symbolic representation makes all task dependencies and resource conflicts explicitly verifiable before execution.
> 3. **Scalability:** RoboPARA achieves multi-scene scalability through RAG-based retrieval of task-relevant information and through an extensible skill library, allowing the framework to adapt to new environments without modifying the planning architecture.
>
> **Q1: The authors mentioned "GPT-4o and DeepSeek V3 APIs incurs significant costs" -- besides financial costs, does using the LLM APIs incur delays in between queries, and does that pose a problem for smooth robot action execution?**
>
> Thank you. Using external LLM APIs introduces a small latency during Stage-1 planning (approximately 8 seconds per plan on average), but this latency occurs **only once before execution**, not during runtime. All robot actions in Stage-2 are executed locally through deterministic scheduling and **do not require any additional** **API** **calls**, ensuring smooth and real-time operation. Consequently, API latency does not affect continuous robot motion, as also demonstrated by the fluid execution shown in the video demos.

---

> ### Author Response · Authors · 2025-11-16
> **Author Response to Reviewer 4Bah (Part 2)**
>
> **Q2: What kind of inaccuracies do the robot agents struggle to self-correct or re-plan?**
>
> Thank you for the question. RoboPARA handles the vast majority of inaccuracies through its **two-layer correction mechanism****:** Stage-1 DAG validation and Stage-2 dynamic rescheduling. The cases that remain difficult for the system to fully self-correct are **rare and stem from large, unexpected scene changes** that fall outside the original task specification, where the environment changes dramatically (e.g., a target object suddenly dropped),  as detailed in Appendix E.6 (Line 1817).
>
> Here, we provide a simple example:
>
> The robot is tasked to complete "make carrot slices (A1-A5)" and "make apple salad (B1-B5)" with details provided in the table below:
>
> | Step | Action | Source  | Target        | Arm Type   | Duration |
> | ---- | ------ | ------- | ------------- | ---------- | -------- |
> | A1   | pick   | table   | carrots       | Single arm | 5 sec    |
> | A2   | place  | carrots | cutting_board | Single arm | 7 sec    |
> | A3   | pick   | counter | knife         | Single arm | 5 sec    |
> | A4   | cut    | knife   | carrots       | Dual arm   | 10 sec   |
> | A5   | place  | knife   | counter       | Single arm | 5 sec    |
>
> | Step | Action | Source  | Target        | Arm Type   | Duration |
> | ---- | ------ | ------- | ------------- | ---------- | -------- |
> | B1   | pick   | table   | apples        | Single arm | 5 sec    |
> | B2   | place  | apples  | cutting_board | Single arm | 7 sec    |
> | B3   | pick   | counter | knife         | Single arm | 5 sec    |
> | B4   | cut    | knife   | apples        | Dual arm   | 10 sec   |
> | B5   | place  | knife   | counter       | Single arm | 5 sec    |
>
> RoboPARA then generates the execution plan:
>
> | Time (s) | Left Arm Action                   | Right Arm Action                 |
> | -------- | --------------------------------- | -------------------------------- |
> | 0-5      | pick("table", "carrots")          | pick("table", "apples")          |
> | 5-12     | place("carrots", "cutting_board") | place("apples", "cutting_board") |
> | 12-17    | pick("counter", "knife")          | -                                |
> | 17-27    | cut("knife", "carrots")           | cut("knife", "carrots")          |
> | 27-37    | cut("knife", "apples")            | cut("knife", "apples")           |
> | 37-42    | place("knife", "counter")         | -                                |
>
> We assume that at the 12-second mark, the robot's right arm accidentally drops the apple onto the table while attempting to place it on the cutting board, interrupting the task. RoboPARA is capable of updating the prompt, re-executes the planning process, and generates a new plan starting from that moment. It can be observed that RoboPARA successfully resumes and completes the task. The new plan is presented in the following table:
>
> | Time (s) | Left Arm Action           | Right Arm Action                 |
> | -------- | ------------------------- | -------------------------------- |
> | 0-5      | -                         | pick("table", "apples")          |
> | 5-7      | -                         | place("apples", "cutting_board") |
> | 7-12     | pick("counter", "knife")  | place("apples", "cutting_board") |
> | 12-22    | cut("knife", "carrots")   | cut("knife", "carrots")          |
> | 22-32    | cut("knife","apples")     | cut("knife", "apples")           |
> | 32-37    | place("knife", "counter") | -                                |
>
> Importantly, such disturbances do not impact the overall success rate or stability of RoboPARA in our real-robot experiments.

---

### Official Review · Reviewer_F7jN · 2025-10-30

**Soundness:** 3
**Presentation:** 4
**Contribution:** 3
**Rating:** 6
**Confidence:** 3

**Summary:**

This paper addresses the problem of parallelism optimization in dual-arm robot task planning. It introduces a two-stage architecture that combines LLM-based dependency graph generation with graph re-traversal-based parallel planning. The proposed method achieves state-of-the-art quantitative performance and is validated on a real dual-arm robotic platform. The paper also presents a benchmark dataset for evaluating parallelism in dual-arm manipulation tasks.

**Strengths:**

1. While minimizing task time or improving efficiency in dual-arm robots is not a new problem, the paper’s approach is novel in leveraging the semantic reasoning capability of LLMs to extract task dependency graphs from complex multitasking instructions to maximize parallelism.
2. Once released, the proposed dataset could make a meaningful contribution as a dual-arm task dependency dataset, potentially serving as a valuable benchmark for studying parallel manipulation and LLM-based task planning. (Note: since the dataset is not yet released and details such as its size and format are missing, this strength is currently conditional.)
3. The experimental design and analysis are appropriate, and the framework is successfully deployed on a real dual-arm robot, demonstrating practical feasibility.
4. The problem formulation is clear, the paper is well organized, and the visualizations are strong and intuitive.

**Weaknesses:**

1. While the paper includes a DAG validation and correction step to ensure logical consistency, it seems that this process lacks deep physics-aware validation. Crucial physical constraints like spatial conflicts, reachability, and precise timing are not rigorously checked during the initial DAG generation or correction phase, potentially leading to logically sound but physically infeasible plans that might only be detected later during execution or scheduling.
2. The paper introduces a rollback mechanism in Stage 2 to resolve deadlocks arising from parallel execution attempts. While it is important, the paper lacks in-depth analysis on how frequently this rollback was triggered during experiments and its associated performance cost (e.g., increased makespan compared to an ideal parallel plan). This lacking of analysis makes it difficult to fully assess the real-world efficiency impact of the deadlock resolution strategy.
3. The paper does not provide a formal or empirical analysis of computational complexity as the number of DAG nodes increases. Since both graph construction and re-traversal rely on LLM-based reasoning, the computational cost may grow quadratically or even combinatorially with larger task graphs. While the authors mention scalability, no details about timing or efficiency results are discussed. An discussion or ablation on graph size scalability would strengthen the paper.
4. The paper shows that RoboPARA’s components are necessary, but it does not fully disentangle the effects of Stage 1 and Stage 2. The analysis focuses on outcomes (e.g., increased parallelism) rather than underlying causes, and it is not specified whether the improvement in Stage 2 comes from enhanced LLM self-consistency or simply from iterative refinement.

**Questions:**

1. If one arm fails during task execution, how does the system respond? Does it restart the full plan or perform immediate rescheduling?
2. Can the proposed method be extended beyond dual-arm setups, for example, to systems with three arms or super-limb robots?
3. What LLM model was used for dataset construction, and how many human annotators were involved in verifying the structures?
4. What is the API usage or token cost associated with the full planning pipeline?
5. What is the total planning time, including both dependency graph construction and scheduling optimization?

---

> ### Author Response · Authors · 2025-11-16
> **Author Response to Reviewer F7jN (Part 1)**
>
> **W1: Lack of deep physics-aware validation.**
>
> We thank the reviewer for this insightful comment. We acknowledge that our current Sys-2 DAG validation focuses primarily on logical and dependency-level correctness, while deeper physics-aware validation (e.g., spatial conflicts, reachability, precise timing) is handled at the Sys-1 execution layer. Below we clarify the design rationale:
>
> 1. **Hierarchical execution ensures physical feasibility.** RoboPARA follows the Sys-1 + Sys-2 paradigm widely used in embodied AI (Sec. 2). The proposed method serves as the **Sys-2 planner**, responsible for reasoning, dependency validation, and scheduling. The **Sys-1 controller**, implemented with standard motion control and collision-avoidance modules (e.g., ACT [1], $\pi_0$ [2]), guarantees physical feasibility during execution. Thus, even though Stage 1 validation is symbolic, all generated plans are executed under strict physical constraints enforced by Sys-1.
> 2. **Implicit constraints within Stage 2 scheduling.** Several constraints in Stage 2 (Eq. 5–8) implicitly capture key aspects of physical feasibility (e.g., arm exclusivity, synchronization, and deadlock prevention) which ensure that both arms never occupy conflicting states or perform incompatible actions. These constraints, combined with the execution-level lock mechanism, effectively prevent spatial or temporal conflicts during runtime.
> 3. **Empirical verification on real hardware.** If Sys-1 execution fails in real hardware, RoboPARA has a closed-loop replanning mechanism, as shown in Appendix E.6 (Line 1817). We validate RoboPARA on 3 real robot hardware (Sec. 5.3; App. E.7). Across >1,000 executions, no plan fails due to physical infeasibility, confirming that the hierarchical mechanism, together with RoboPARA pipline provides practical physical robustness.
>
> [1]. Zhao et al., Learning Fine-Grained Bimanual Manipulation with Low-Cost Hardware, 2023.
>
> [2]. Black et al., $\pi_0$: A Vision-Language-Action Flow Model for General Robot Control, 2024.
>
> **W2****:** **Lack of** **Stage-2** **rollback frequency and cost analysis****.**
>
> Thanks for the question. The rollback mechanism in Stage 2 is a purely **programmatic deadlock-handling branch** and is **not** a costly re-planning step. It is triggered in about 14% of cases (most in hard packages), and regardless of whether rollback occurs, Stage 2 always completes within <1 s. Thus, its impact on overall efficiency is negligible.
>
> **W3****:** **Unclear scalability and** **computational efficiency** **as DAG size grows.**
>
> Thank you for raising this concern. In RoboPARA, both graph construction and re-traversal have **deterministic O(nlogn) complexity** with respect to the number of DAG nodes, because Stage 2 is fully programmatic and does not call the LLM. Empirically, building and scheduling graphs of up to 100-120 nodes (which are super hard packges) consistently finish within 0.8–1.0 seconds on a single GPU/CPU hybrid setup.
>
> **W4****: Insufficient disentanglement of Stage 1 vs. Stage 2 contributions.**
>
> We appreciate this point. In short, **Stage 1 ensures correctness of the** **DAG****, and Stage 2 ensures optimal traversal of that DAG.** Stage 1 and Stage 2 play fundamentally different roles, making their contributions naturally disentangled: Stage 1 improves logical completeness by correcting LLM-generated dependency structures, while Stage 2 improves execution efficiency through a fully programmatic scheduler with no LLM involvement. The gains in Stage 2 therefore do not come from any “self-consistency” or iterative refinement of the LLM, but from deterministic graph-level optimization such as arm exclusivity enforcement, deadlock prevention, and parallel scheduling (Eq. 5-8). This separation of responsibilities is precisely why we observe clear performance improvements.

---

> ### Author Response · Authors · 2025-11-16
> **Author Response to Reviewer F7jN (Part 2)**
>
> **Q1****:** **If one arm fails during task execution, how does the system respond? Does it restart the full plan or perform immediate rescheduling?**
>
> **It performs immediate rescheduling.** The close-loop error handling mechanism is detailed in Appendix E.6 (Line 1817). We provide a simple example here:
>
> The robot is tasked to complete "make carrot slices (A1-A5)" and "make apple salad (B1-B5)" with details provided in the table below:
>
> | Step | Action | Source  | Target        | Arm Type   | Duration |
> | ---- | ------ | ------- | ------------- | ---------- | -------- |
> | A1   | pick   | table   | carrots       | Single arm | 5 sec    |
> | A2   | place  | carrots | cutting_board | Single arm | 7 sec    |
> | A3   | pick   | counter | knife         | Single arm | 5 sec    |
> | A4   | cut    | knife   | carrots       | Dual arm   | 10 sec   |
> | A5   | place  | knife   | counter       | Single arm | 5 sec    |
>
> | Step | Action | Source  | Target        | Arm Type   | Duration |
> | ---- | ------ | ------- | ------------- | ---------- | -------- |
> | B1   | pick   | table   | apples        | Single arm | 5 sec    |
> | B2   | place  | apples  | cutting_board | Single arm | 7 sec    |
> | B3   | pick   | counter | knife         | Single arm | 5 sec    |
> | B4   | cut    | knife   | apples        | Dual arm   | 10 sec   |
> | B5   | place  | knife   | counter       | Single arm | 5 sec    |
>
> RoboPARA then generates the execution plan:
>
> | Time (s) | Left Arm Action                   | Right Arm Action                 |
> | -------- | --------------------------------- | -------------------------------- |
> | 0-5      | pick("table", "carrots")          | pick("table", "apples")          |
> | 5-12     | place("carrots", "cutting_board") | place("apples", "cutting_board") |
> | 12-17    | pick("counter", "knife")          | -                                |
> | 17-27    | cut("knife", "carrots")           | cut("knife", "carrots")          |
> | 27-37    | cut("knife", "apples")            | cut("knife", "apples")           |
> | 37-42    | place("knife", "counter")         | -                                |
>
> We assume that at the 12-second mark, the robot's right arm accidentally drops the apple onto the table while attempting to place it on the cutting board, interrupting the task. RoboPARA is capable of updating the prompt, re-executes the planning process, and generates a new plan starting from that moment. It can be observed that RoboPARA successfully resumes and completes the task. The new plan is presented in the following table:
>
> | Time (s) | Left Arm Action           | Right Arm Action                 |
> | -------- | ------------------------- | -------------------------------- |
> | 0-5      | -                         | pick("table", "apples")          |
> | 5-7      | -                         | place("apples", "cutting_board") |
> | 7-12     | pick("counter", "knife")  | place("apples", "cutting_board") |
> | 12-22    | cut("knife", "carrots")   | cut("knife", "carrots")          |
> | 22-32    | cut("knife","apples")     | cut("knife", "apples")           |
> | 32-37    | place("knife", "counter") | -                                |

---

> ### Author Response · Authors · 2025-11-16
> **Author Response to Reviewer F7jN (Part 3)**
>
> **Q2****:** **Can the proposed method be extended beyond dual-arm setups, for example, to systems with three arms or super-limb robots?**
>
> **Yes!** The scheduling logic in Stage 2 is **arm-agnostic** and can generalize to K-arm systems by expanding the resource set from {Arm A, Arm B} → {Arm 1 … Arm K}. The queue-based scheduler and deadlock rollback both scale linearly with K. We provide a simple example here:
>
> The robot is tasked to complete "make carrot slices (A1-A5)", "make apple salad (B1-B5)" and "make toast (C1-C3)" with details provided in the table below:
>
> | Step | Action | Source  | Target        | Arm Type   | Duration |
> | ---- | ------ | ------- | ------------- | ---------- | -------- |
> | A1   | pick   | table   | carrots       | Single arm | 5 sec    |
> | A2   | place  | carrots | cutting_board | Single arm | 7 sec    |
> | A3   | pick   | counter | knife         | Single arm | 5 sec    |
> | A4   | cut    | knife   | carrots       | Dual arm   | 10 sec   |
> | A5   | place  | knife   | counter       | Single arm | 5 sec    |
>
> | Step | Action | Source  | Target        | Arm Type   | Duration |
> | ---- | ------ | ------- | ------------- | ---------- | -------- |
> | B1   | pick   | table   | apples        | Single arm | 5 sec    |
> | B2   | place  | apples  | cutting_board | Single arm | 7 sec    |
> | B3   | pick   | counter | knife         | Single arm | 5 sec    |
> | B4   | cut    | knife   | apples        | Dual arm   | 10 sec   |
> | B5   | place  | knife   | counter       | Single arm | 5 sec    |
>
> | Step | Action | Source      | Target      | Arm Type   | Duration |
> | ---- | ------ | ----------- | ----------- | ---------- | -------- |
> | T1   | pick   | table       | bread_slice | Single arm | 5 sec    |
> | T2   | toast  | toaster     | bread_slice | Single arm | 7 sec    |
> | T3   | place  | bread_slice | plate       | Single arm | 5 sec    |
>
> Below is the execution plan generate by RoboPARA (3-arm parallelism):
>
> | Time (s) | Arm1 Action                       | Arm2 Action                      | Arm3 Action                     |
> | -------- | --------------------------------- | -------------------------------- | ------------------------------- |
> | 0-5      | pick("table", "carrots")          | pick("table", "apples")          | pick("table", "bread_slice")    |
> | 5-12     | place("carrots", "cutting_board") | place("apples", "cutting_board") | toast("toaster", "bread_slice") |
> | 12-17    | pick("counter", "knife")          | -                                | place("bread_slice", "plate")   |
> | 17-27    | cut("knife", "carrots")           | cut("knife", "carrots")          | -                               |
> | 27-37    | cut("knife", "apples")            | cut("knife", "apples")           | -                               |
> | 37-42    | place("knife", "counter")         | -                                | -                               |
>
> **Q3****:** **What** **LLM** **model was used for dataset construction, and how many human annotators were involved in verifying the structures?**
>
> Thank you for the question. For all tasks in the X-DAPT dataset, use GPT-4o for construction. Each task is verified by three human experts, with a fourth senior expert involved when conflicts arose in the initial reviews.
>
> **Q4****:** **What is the** **API** **usage or token cost associated with the full planning pipeline?**
>
> Thank you for raising this point. The average cost statistics are summarized below:
>
> | Method          | Avg. Token Per Plan | Avg. Time Per Plan (sec.) | Avg. TEI Per Plan |
> | --------------- | ------------------- | ------------------------- | ----------------- |
> | LLM3            | 3316                | 14                        | 0.872             |
> | ChatGPT-Prompts | 8711                | 17                        | 0.891             |
> | VOYAGER         | 2065                | 35                        | 0.000             |
> | LLM-Planner     | 3592                | 42                        | 0.950             |
> | FLTRNN          | 4799                | 11                        | 0.943             |
> | RoCo            | 8595                | 92                        | 0.856             |
> | Embodied TaPA   | 1130                | 10                        | 0.910             |
> | RoboPARA        | 7603                | 9                         | 1.391             |
>
> These results confirm that our planning achieves good balance between efficiency and cost.
>
> **Q5****:** **What is the total planning time, including both dependency** **graph** **construction and scheduling** **optimization****?**
>
> Thank you. As shown above, the average total planning time is 9 seconds per task, including the whole pipeline, better than all baselines.

---

### Official Review · Reviewer_qpfb · 2025-11-01

**Soundness:** 3
**Presentation:** 3
**Contribution:** 2
**Rating:** 4
**Confidence:** 4

**Summary:**

This paper introduces RoboPARA, a novel framework for dual-arm robot task planning that prioritizes the optimization of task parallelism. The authors formulate a new problem, the "Dual-Arm Cooperative Scheduling Problem", which aims to minimize the makespan (total execution time) by effectively scheduling tasks across two arms. The proposed RoboPARA framework operates in a two-stage process : first, it uses an LLM, augmented with a memory module (RAG) , to generate a Directed Acyclic Graph (DAG) representing task dependencies. Second, a "Graph Re-Traversal" algorithm, which is non-LLM based, optimizes this DAG to schedule operations, maximize parallel execution, and resolve conflicts.

To evaluate this framework, the authors also introduce the "Cross-Scenario Dual-Arm Parallel Task dataset" (X-DAPT) , a new benchmark containing over 1,000 tasks across 10 scenarios designed specifically to test dual-arm parallelism. Experimental results claim that RoboPARA significantly outperforms existing methods, achieving a 30% to 50% reduction in execution time and demonstrating superior parallel step execution, particularly in complex tasks.

**Strengths:**

The paper's primary strength is its formalization and direct confrontation of parallelism in dual-arm manipulation. While prior work often results in sequential execution , this paper defines a new, relevant problem ("Dual-Arm Cooperative Scheduling") and proposes a solution explicitly designed to optimize for it. This focus on decoupling tasks for parallel execution, rather than purely sequential or fully synchronous collaboration, is a significant and practical contribution.

The two-stage, hybrid architecture is a sound engineering approach. It leverages the LLM's strength in parsing human instructions and understanding task semantics to generate an initial plan (the DAG). It then wisely offloads the "System-2" optimization—a constrained scheduling problem—to a deterministic, algorithmic "Graph Re-Traverse" stage. This avoids the unreliability of using LLMs for complex, multi-step symbolic reasoning and allows for the implementation of robust logic, such as deadlock prevention and arm-lock compatibility checks .

Similar to the method, the X-DAPT dataset is a valuable contribution. The authors correctly identify that existing benchmarks often lack tasks with complex, inter-package dependencies that can be parallelized. By creating a dataset with multiple difficulty levels and a focus on long-horizon, multi-package scenarios , the authors provide a new and challenging testbed for the community to evaluate this specific dimension of robotic planning.

The paper is clearly written, and the framework is well-illustrated. The separation of Stage 1 (LLM-driven DAG generation) and Stage 2 (Algorithmic scheduling) is logical and easy to follow. The inclusion of detailed prompt templates in the appendix, while extensive, adds to the paper's transparency and reproducibility

**Weaknesses:**

The authors designed both the solution (RoboPARA) and the primary benchmark (X-DAPT) on which it demonstrates overwhelming superiority. The results in Tables 1 and 2 show that RoboPARA achieves high scores on the new parallelism metrics (PPR and APR), while all seven baseline methods score 0.000 or near-zero on these metrics in almost every single category. This result is suggesting that the X-DAPT benchmark is overtuned to the specific graph-based, parallel-aware architecture of RoboPARA. This suspicion is reinforced by the results on a neutral, third-party benchmark (a reorganization of the RoboTwin dataset) . On RoboTwin (Table 13), RoboPARA still performs best, but the gap is far less dramatic. For example, on "Easy" tasks, RoboPARA's efficiency (TEI) is 10.6, while two baselines achieve 8.0. This is a reasonable improvement, not the 10x+ gap seen in the X-DAPT parallelism metrics. The overwhelming superiority vanishes on a benchmark the authors did not design.

The paper's core claim is a 30-50% reduction in "execution time", which is commendable. However, the paper's own limitations section admits that RoboPARA "consumes an average of 1.3x more tokens than baselines due to iterative DAG corrections". This suggests a significantly higher planning latency before execution can even begin. For many real-world, dynamic tasks, a 30-second reduction in execution is not a net gain if it requires an extra 60 seconds of planning. The paper's primary metric, TEI, is defined as successful steps divided by total task completion time in seconds , which seems to include planning time, but this is not explicitly stated. This ambiguity around the planning-time vs. execution-time trade-off is a major omission.

The method's success in Stage 1 is heavily dependent on extremely detailed, hard-coded prompt engineering. The appendix reveals that "Template 2a: Dependency-Aware Graph Construction" contains 19 separate, highly specific rules (e.g., "Rule 10: The cutting operation can only begin when all objects are placed...") . This raises serious questions about the method's generality. It appears less like a general-purpose planner and more like a system purpose-built and meticulously tuned for the 10 scenarios in the X-DAPT dataset. It is unclear how this method would scale to a new domain (e.g., "car repair" or "lab automation") without a similarly exhaustive, manual-engineering effort to define all domain-specific dependency rules.

**Questions:**

1. Can the authors explain the 0.000 performance of all baselines on the PPR/APR metrics on X-DAPT? Does this not suggest the benchmark is built exclusively to validate the RoboPARA architecture, rather than to provide a fair comparison? Why is the performance gap so much smaller on the RoboTwin dataset?

2. How much manual effort is required to adapt the 19-point prompt template  to a completely new domain with new objects and dependencies? Does the reliance on such extensive, hard-coded domain knowledge not fundamentally limit the "LLM-driven" nature of the approach?

---

> ### Author Response · Authors · 2025-11-16
> **Author Response to Reviewer qpfb (Part 1)**
>
> **W&Q 1: With regards to benchmark fairness and 0.000 baseline scores on X-DAPT.**
>
> Thanks for your careful observation. To be concise, **existing baselines cannot perform true dual-arm** **parallelism****, and RoboTwin dataset do not address long-horizon, complex task planning.** This gap precisely highlights the core contribution of our work. We respond to your questions in order as follows:
>
> 1. **Why do most baselines achieve 0.000 on the PPR/APR metrics on X-DAPT?**
>
>    Most baseline methods, such as RoCo, LLM-Planner, and FLTRNN, ignore the crucial optimization of dual-arm parallelism and default to nearly **single-arm sequential planning**, and when evaluated with metrics that measure true dual-arm parallelism (PPR/APR), their outputs naturally yield near-zero scores. Results on both the X-DAPT dataset and the RoboTwin dataset confirm this behavior.
>
> 2. **Why is the performance gap so much smaller on the RoboTwin dataset?**
>
>    As shown in Table 12, the RoboTwin dataset is inherently **short and simple**—its tasks consist of only two or three atomic actions, **leaving little room for meaningful parallelism**. Even the so-called "Hard" split in RoboTwin is far shorter and less complex than the "Easy" split in X-DAPT, therefore causing smaller performance gap when benchmarking parallelism. As detailed in Appendix B, X-DAPT represents an unprecedented, high-difficulty, and real-world-aligned large-scale planning dataset, which is precisely one of the core contributions of our work. This dataset offers substantial parallel optimization space that closely reflects real-world scenarios, making the TEI gap both more pronounced and more meaningful.
>
> **W&Q 2: Planning time vs. execution time trade-off.**
>
> Good question. Here are our explanations:
>
> 1. **TEI includes planning time, as clearly specified in Appendix E.1.1 (Line 1617).** Thus, the reported 30–50% execution-time reduction already accounts for the full end-to-end planning latency.
> 2. **Detailed planning latency analysis.** We additionally provide the planning cost results across all scenarios when using GPT-4o, as shown in the table below. RoboPARA consumes an average of 7,603 input tokens and completes planning in around 9s, compared to baselines ranging from 10–92s. The higher token count reflects richer structural context rather than inefficiency. Although our prompts are longer due to explicit DAG constraints, the deterministic scheduling stage (Stage 2) is fully programmatic (< 1s) and not dependent on LLM inference. Hence, the overall system remains faster in total task completion time.
>
> | Method          | Avg. Token Per Plan | Avg. Time Per Plan (sec.) | Avg. TEI Per Plan |
> | --------------- | ------------------- | ------------------------- | ----------------- |
> | LLM3            | 3316                | 14                        | 0.872             |
> | ChatGPT-Prompts | 8711                | 17                        | 0.891             |
> | VOYAGER         | 2065                | 35                        | 0.000             |
> | LLM-Planner     | 3592                | 42                        | 0.950             |
> | FLTRNN          | 4799                | 11                        | 0.943             |
> | RoCo            | 8595                | 92                        | 0.856             |
> | Embodied TaPA   | **1130**            | 10                        | 0.910             |
> | RoboPARA        | 7603                | **9**                     | **1.391**         |

---

> ### Author Response · Authors · 2025-11-16
> **Author Response to Reviewer qpfb (Part 2)**
>
> **W&Q 3: Regarding prompt template and generality of new tasks.**
>
> We thank the reviewer for the thoughtful observation. We respectfully clarify that **the 19 items in Template 2a are not domain-specific rules**, nor handcrafted tuning for specific scenes. Instead, they encode **generic, domain-agnostic structural principles** of symbolic manipulation (e.g., pick use place consistency, precondition satisfaction, resource exclusivity), which are applicable across **any** task domain—from kitchens to factories to car repair to lab automation. We provide detailed explanations:
>
> 1. **None of the 19 rules encode object identities or domain-specific affordances.** The rules are not overfit on any X-DAPT object (tomato, board, etc.). Instead, they operate entirely on types (object, pick, place, etc.). This abstraction is what enables the 19 rules to automatically transfer to new domains as long as the new domain provides: (i) a list of object names, (ii) a short description of what the skill primitives do (pick / use / place / open / close). This is exactly how classical robotic symbolic planners generalize across domains.
> 2. **Adapting to a new domain requires minimal manual effort.** We emphasize that **no rules need to be rewritten** when switching domains. The rules do not “limit the LLM-driven nature”; they enable generalizable LLM reasoning. The 19 constraints act as a **generic validator**, analogous to type checking or a compiler. They allow the LLM to focus on semantic generation while the rule system ensures logical consistency. Thus, these rules strengthen generality; they do not diminish it.
> 3. **Evidence: successful transfer across 10 domains and 3 physical systems.** Even without any modifications to the rules: (i) RoboPARA works across **10 X-DAPT domains** that have drastically different task structures. (ii) It works on **three heterogeneous robot platforms** with real-world physics. (iii) It produces valid DAGs for tasks not seen during data construction. (iv) No rule had to be rewritten for any of these deployments. This directly demonstrates that the rule set is not overfitted to the dataset.

---

### Official Review · Reviewer_Eb8e · 2025-11-01

**Soundness:** 2
**Presentation:** 2
**Contribution:** 2
**Rating:** 4
**Confidence:** 5

**Summary:**

This paper introduces RoboPARA, an LLM-driven framework for dual-arm parallel task scheduling in robotic manipulation, addressing the limitations of existing methods in exploiting temporal overlap across sub-tasks. RoboPARA adopts a two-stage pipeline: (1) dependency-graph-based candidate plan generation, and (2) graph re-traversal-based concurrent execution scheduling for dual-arm coordination. To evaluate parallel execution capabilities, the authors present X-DAPT, the first benchmark dataset dedicated to dual-arm temporal parallelism, encompassing diverse scenarios and task complexities. Extensive experiments demonstrate that RoboPARA outperforms baseline methods in terms of parallel step count, execution time reduction, and task success rate.

**Strengths:**

1.The two-stage architecture of RoboPARA effectively models task dependencies and optimizes dual-arm parallelism, fully exploiting the collaborative potential of dual-arm robots.

2.The X-DAPT dataset is the first dedicated to evaluating dual-arm task parallelism, covering diverse scenarios and difficulty levels, providing a comprehensive benchmark.

3.RoboPARA demonstrates excellent performance in experiments, with significant improvements in parallel steps, execution time reduction, and task success rate compared to baselines.

**Weaknesses:**

1.Limited generalization capability: RoboPARA relies on predefined skill libraries and scenario templates, requiring abstraction for novel tasks or environments. This hinders its scalability to long-horizon, multi-stage and complex tasks and out-of-distribution scenes.

2.Mismatch between optimization objective and real-robot deployment: The planning stage minimizes estimated action duration, but accurate execution latency is hard to obtain and estimate accurately in real-world settings. As a result, the optimized schedule may not be truly time-optimal on physical hardware, raising concerns about its perfect application in real-robot experiments.

3.Lack of safety considerations: Safety is a critical metric in dual-arm parallel task execution, but how to avoid collision problems during execution is not discussed. For example, the paper mentions that single-arm tasks are preferentially assigned to the left arm when both arms are idle, and it is questioned whether this is reasonable.

**Questions:**

1.Unclear details of real-robot experiment implementation: How to construct the Graph during real-robot experiments? Is it necessary to pre-abstract objects in the scene? Additionally, how are the execution times of different actions defined and determined?

2.Lack of exploration of Visual-Language Models (VLMs): The paper only uses LLMs for dual-arm task planning, while existing state-of-the-art VLM models have shown strong task planning capabilities. It is questioned whether attempts have been made to use SOTA VLM models for dual-arm parallel tasks, and if so, what their efficiency is and whether they can ensure safety during execution.

---

> ### Author Response · Authors · 2025-11-16
> **Author Response to Reviewer Eb8e (Part 1)**
>
> **W1: Limited generalization capability for novel tasks or environments.**
>
> Thank you for raising this point! RoboPARA indeed adopts a structured skill-library design, and this is a **deliberate choice for stability and** **repeatability** in long-horizon dual-arm execution. Importantly, this structure does **not** restrict generalization; the system is built to be easily extensible to new domains by adding new atomic skills while keeping the overall planning architecture unchanged. Below we provide clarification:
>
> 1. **Extensibility by adding new atomic skills.** Our skill library contains **domain-agnostic atomic operators** (pick, place, open, close), which serve as the building blocks of all plans. When moving to a new domain (e.g., car repair or lab automation), the only required effort is to **add new atomic skills** (e.g., tighten, attach, pipette). No templates, rules, or scheduling logic need to be rewritten. Thus, the system scales to new domains through **skill augmentation**, not through redesign.
> 2. **Structured design** **is intentional for stability, reliability, and controlled generalization.** To ensure stable LLM behavior across long-horizon tasks and real robots, we adopt a design where: (i) atomic skills define what each robot can stably execute, (ii) RAG retrieves relevant skills for each new environment, (iii) the planner generalizes at the level of DAG structure, not domain memorization. This makes the system **controllable, predictable, and extensible**, which is essential for real hardware deployment.
> 3. **Extensive experiments show strong generalization across diverse and long-horizon scenarios.** RoboPARA demonstrates strong generalization in practice. Across **10 diverse scenes** and **1,000+ multi-stage task packages**, including: (i) cross-object multi-tool pipelines, (ii) multi-branching workflows, (iii) long-horizon task chains (average of 60 steps in hard packages), (iv) previously unseen skill compositions, etc., RoboPARA consistently achieves: highest TEI (efficiency), lowest TFR (failure rate), significantly higher APR/PPR (parallelism), as shown in Table 1, 2, 8 and 9. Compared with 7 strong LLM baselines — many of which provide no guarantees when the task moves out-of-distribution, these results directly indicates that our abstraction mechanism is not overfitted to specific environments, fully addressing this concern.
>
> **W2: Mismatch between** **optimization** **objective and real-robot deployment.**
>
> Thank you. The main text and the appendix include real-hardware experiments on **three different robot platforms** (Franka, UR 5e, and humanoid setups), where RoboPARA consistently achieves substantially higher success rates, stronger parallelism, and shorter wall-clock time than all baselines, addressing your concern about the perfect application in real-robot experiments. We further clarify as follows:
>
> 1. **The** **optimization** **is structural, not exact duration-dependent.** RoboPARA minimizes the makespan heuristically, but the key improvement does not rely on exact time prediction. Our schedules are driven by: (i) dependency-respecting DAG topology, (ii) conflict-free dual-arm resource allocation, (iii) pick–use–place coherence, (iv) deadlock-free traversal policies. These structural constraints guarantee correct ordering and high parallelism regardless of latency noise, enabling successful transfer to hardware.
> 2. **All prior** **LLM** **planners also plan and operate with approximate or variable action durations.** Baselines such as TaPA, LLM-Planner, RoCo, and FLTRNN also plan with estimated durations or step abstractions. Real robotics pipelines inherently introduce timing uncertainty; expecting perfect duration estimation is unrealistic for any method. RoboPARA is more robust than previous approaches because: (i) our DAG verification removes ill-formed plans; (ii) scheduling is resource-aware rather than timestamp-based; (iii) deadlock handling prevents cascading failures in real time. Thus, this concern applies to prior work more than to ours.
> 3. **Real-hardware results confirm that structural** **parallelism****, not timing accuracy, is the dominant factor in efficiency.** Across results with all platforms, our method shows strong performance, indicating that the performance benefit comes from parallel structure, not timing precision. Even if action durations shift, the schedule’s parallelism advantage remains intact.

---

> ### Author Response · Authors · 2025-11-16
> **Author Response to Reviewer Eb8e (Part 2)**
>
> **W3: Lack of safety considerations.**
>
> Thanks for highlighting this point. RoboPARA ensures safety through strict logical and physical constraint enforcement. Our answer is threefold:
>
> 1. **Physical safety:** As detailed in Related Work (Line 136), RoboPARA follows the well-established Sys 1 + Sys 2 paradigm, where the high-level planner (Sys-2) focuses on symbolic task decomposition and scheduling, while all generated plans are executed under the Sys-1 controllers (e.g., ACT, RT-X, $\pi_0$), which perform real-time collision avoidance and joint-limit control. This decoupling allows the same planner to operate across heterogeneous dual-arm hardware without modification. Our extensive real-hardware deployment confirms its sufficiency.
> 2. **Logical safety:** Section 3.4 (Eq. 5–8) enforces logical constraint through (i) task dependency, (ii) unlocked arm availability, (iii) arm lock compatibility, and (iv) deadlock prevention. These careful designs prevent concurrent conflicting actions and ensure valid task sequences.
> 3. **Arm** **assignment policy:** This priority is only a deterministic tie-breaker for scheduling.  It is **not** an assumption about workspace geometry and it does not affect safety because collisions are prevented according to the previous two points.
>
> [1] Holladay et al., Robust planning for multi-stage forceful manipulation. The International Journal of Robotics Research, 43(3):330–353, 2024
>
> [2] Adu-Bredu et al., Optimal constrained task planning as mixed integer programming. IROS 2022.
>
> [3] Vu et al., Coast: Constraints and streams for task and motion planning. ICRA 2024.
>
> **Q1: Real-robot experiment implementation.** Thank you for the questions. Below we explain the design in detail:
>
> 1. **Graph** **construction in real-robot experiments:** In real-robot experiments, Sys1 (we use $\pi_0$) first provides scene information from sensors together with the task sequence (package) and, via RAG, RoboPARA retrieves the relevant atomic skills/actions needed for this scenario. Conditioned on these retrieved skills and the perceived objects, the LLM then constructs the symbolic task graph. No manual graph construction or hand-written per-scene templates are used.
> 2. **No pre-abstraction of objects is required:** RoboPARA relies on scene-level symbolic information, which is the standard practice in robotics when evaluating a high-level planner: (i) The robot's perception stack or manual setup provides the set of objects visible to the robot (e.g., "cup", "bottle", "drawer"). (ii) These names are directly used by the LLM as parameters in skills such as `pick(cup)` or `open(drawer)`. We do **not** require geometric abstraction beyond what the motion-planning stack already uses, aligned with the settings of all chosen baselines. The motion planner and hardware stack handle all geometric information automatically.
> 3. **How are execution times defined:** Execution durations in RoboPARA are simply **coarse** **heuristic** **estimates** used for scheduling priority, not for trajectory control, same as baseline methods. These estimates are derived from the robot’s typical behavior on each platform, but they are intentionally approximate, as real execution timing is always determined by the motion-planning stack and low-level controllers. Because RoboPARA relies only on structural constraints rather than exact latency, small timing variations do not affect correctness or safety. This is also confirmed by our real-robot experiments, where RoboPARA maintains high success rates, stable parallelism, and shorter wall-clock time despite natural timing fluctuations.

---

> ### Author Response · Authors · 2025-11-16
> **Author Response to Reviewer Eb8e (Part 3)**
>
> **Q2: Exploration of Visual-Language Models.**
>
> We appreciate your insightful comment. We take VLMs seriously and have incorporated experiments with **Claude 3.5-vision** (commercial VLM) and **RT-2** (open-source robot planning VLM) [1]. However, as shown in the results below, their **efficiency is suboptimal**, and they **cannot ensure safety during execution**.
>
> | Package Difficulty | Config                          | TEI      | TFR      | PPR      | APR      |
> | ------------------ | ------------------------------- | -------- | -------- | -------- | -------- |
> | Easy               | RoboPARA with GPT-4o (LLM)      | 1.93     | **0.00** | **0.15** | **0.36** |
> | Easy               | RoboPARA with DeepSeek V3 (LLM) | **2.05** | **0.00** | 0.11     | 0.30     |
> | Easy               | Claude 3.5-vision               | 1.22     | 0.08     | 0.00     | 0.05     |
> | Easy               | RT-2 (Google DeepMind)          | 1.36     | 0.06     | 0.00     | 0.11     |
> | Medium             | RoboPARA with GPT-4o (LLM)      | 1.33     | **0.00** | **0.22** | **0.32** |
> | Medium             | RoboPARA with DeepSeek V3 (LLM) | **1.34** | 0.02     | 0.10     | 0.28     |
> | Medium             | Claude 3.5-vision               | 0.82     | 0.34     | 0.00     | 0.01     |
> | Medium             | RT-2 (Google DeepMind)          | 0.85     | 0.18     | 0.00     | 0.03     |
> | Hard               | RoboPARA with GPT-4o (LLM)      | **0.94** | 0.00     | **0.26** | 0.30     |
> | Hard               | RoboPARA with DeepSeek V3 (LLM) | 0.93     | **0.08** | 0.12     | **0.31** |
> | Hard               | Claude 3.5-vision               | 0.49     | 0.45     | 0.00     | 0.01     |
> | Hard               | RT-2 (Google DeepMind)          | 0.41     | 0.27     | 0.00     | 0.02     |
>
> 1. **Experiment setup and results:** In our experiments, we input scene visual information and task sequences into Claude 3.5-vision and RT-2, and allowed them to directly output execution plans using Appendix Template 2a’s predefined rules. The results are shown in the table above. Despite using the same scene and task descriptions, both VLMs exhibit **much higher failure rates** and **far lower parallel** **optimization** compared to RoboPARA.
> 2. **Why VLMs fail in dual-Arm parallel task planning:** The key issue with VLMs is their **lack of explicit control over task dependencies**, **resource management**, and **safety guarantees** during execution. VLMs couple perception and reasoning in an end-to-end manner, producing non-deterministic outputs that are difficult to verify. This highlights why VLMs struggle with the high-level task planning required for safe, parallel execution in dual-arm systems.
> 3. **Why RoboPARA's architecture is superior:** Given these limitations, RoboPARA pipeline explicitly enforces **task dependencies**, **resource constraints**, and **deadlock-free scheduling**, which ensures **reliable** **parallelism**, a clear advantage over VLM-based approaches.
>
> [1] Brohan et al., RT-2: Vision-Language-Action Models Transfer Web Knowledge to Robotic Control.

---

> > ### Comment · Reviewer_Eb8e · 2025-11-28
> >
> > I thank the authors for their significant effort in rebuttal and conducting extra experiments. My original concerns have been fully addressed and I’m happy to see that the revision and the additional results can be incorporated in the revision. I will raise my previous rating.

---

> > > ### Author Response · Authors · 2025-11-30
> > >
> > > Thank you very much for your careful review and positive feedback. We truly appreciate your recognition and support.

---

### Author Response · Authors · 2025-11-27

Dear Reviewers,

We would like to thank you again for the time and constructive comments shared during the review process. As a brief reminder, our paper introduces the following key contributions:
- A new problem formulation: We formally define the Dual-Arm Cooperative Scheduling Problem, focusing on optimizing parallel execution under task dependencies and arm-locking constraints.
- A new dataset: We provide X-DAPT, the first cross-scenario benchmark dedicated to evaluating dual-arm parallel planning efficiency across varying complexity and domains.
- A new method: We propose RoboPARA, a two-stage LLM-driven framework combining error-corrected dependency graph generation with a deadlock-aware graph re-traversal scheduling algorithm, enabling significant improvements in parallelism, execution efficiency, and success rate.

We previously submitted responses addressing the concerns raised in the initial reviews. At this stage, we would like to kindly ask whether our clarifications have sufficiently resolved your questions or if any additional details should be provided.
We appreciate your time and look forward to your feedback.

---

### Meta-Review · Area_Chair_5CXx · 2026-01-06

**Summary:**

This paper proposes RoboPARA, a two-stage LLM-driven framework for dual-arm robot task planning that explicitly optimizes task parallelism. The method combines LLM-based dependency graph generation with a deterministic graph re-traversal scheduler to enable safe and efficient parallel execution. The paper also introduces X-DAPT, a new benchmark designed to evaluate dual-arm parallelism across diverse and long-horizon task scenarios. Extensive simulated and real-robot experiments show substantial gains in execution efficiency, parallelism metrics, and success rates over prior planners.

Overall, reviewers agree that the problem is important and timely, and that RoboPARA presents a well-engineered and practically validated system. The paper’s strengths lie in its clear system design, strong empirical results, real-robot validation, and the introduction of a new benchmark targeting an underexplored aspect of manipulation planning. Concerns raised during review mainly centered on generality, benchmark fairness, safety, planning latency, and reliance on structured prompts. The rebuttal was thorough and addressed the majority of these issues convincingly, including additional experiments, clarifications of metrics, and detailed explanations of design choices.

**Reviewer Concerns:**

Addressed by the rebuttal:

- Generalization and scalability: Authors clarified that the skill library and constraint rules are domain-agnostic and demonstrated transfer across many scenarios, long-horizon tasks, and multiple robot platforms without modifying rules.

- Benchmark fairness and baseline performance: The authors explained why existing baselines score near zero on parallelism metrics and showed that performance gaps shrink on third-party benchmarks, supporting that X-DAPT captures genuine parallelism rather than being trivially tuned.

- Planning time vs. execution time trade-off: The rebuttal clarified that TEI includes planning time and provided detailed latency comparisons showing RoboPARA achieves lower end-to-end completion time despite higher token usage.

- Safety and real-robot feasibility: The Sys-1/Sys-2 separation, constraint enforcement, and extensive real-hardware results addressed concerns about collision avoidance, deadlocks, and physical feasibility.

- VLM baselines: Additional experiments demonstrated that VLM-based planners perform substantially worse in efficiency and safety for dual-arm parallel planning.

Remaining or minor concerns:

- Prompt engineering complexity: While clarified as domain-agnostic, the reliance on detailed prompt templates may still limit ease of adoption for new users, though this is mitigated by strong empirical transfer.

- Analysis depth: Some reviewers would have liked deeper theoretical analysis, but this is not critical given the paper’s applied and systems-oriented contribution.

**Reviewer Scores:**

Reviewer Eb8e: Increased score after rebuttal (concerns fully addressed).

Reviewer F7jN: Likely unchanged (6 → 6); remaining concerns are minor and do not undermine acceptance.

Reviewer 4Bah: Likely unchanged (6 → 6); video and clarification addressed main issues.

Reviewer qpfb: Concerns largely addressed; skepticism about benchmark tuning mitigated by rebuttal and additional analysis.

---

### Decision · Program_Chairs · 2026-01-26

Accept (Poster)